# Application of 3D MAPs pipeline identifies the morphological sequence chondrocytes undergo and the regulatory role of GDF5 in this process

Sarah Rubin [1], Ankit Agrawal[1], Johannes Stegmaier[2,3], Sharon Krief[1], Neta Felsenthal [1], Jonathan Svorai[1], Yoseph Addadi [4], Paul Villoutreix [5✉], Tomer Stern [1✉] & Elazar Zelzer [1✉]

The activity of epiphyseal growth plates, which drives long bone elongation, depends on extensive changes in chondrocyte size and shape during differentiation. Here, we develop a pipeline called 3D Morphometric Analysis for Phenotypic significance (3D MAPs), which combines light-sheet microscopy, segmentation algorithms and 3D morphometric analysis to characterize morphogenetic cellular behaviors while maintaining the spatial context of the growth plate. Using 3D MAPs, we create a 3D image database of hundreds of thousands of chondrocytes. Analysis reveals broad repertoire of morphological changes, growth strategies and cell organizations during differentiation. Moreover, identifying a reduction in Smad 1/5/9 activity together with multiple abnormalities in cell growth, shape and organization provides an explanation for the shortening of *Gdf5* KO tibias. Overall, our findings provide insight into the morphological sequence that chondrocytes undergo during differentiation and highlight the ability of 3D MAPs to uncover cellular mechanisms that may regulate this process.

[1] Department of Molecular Genetics, Weizmann Institute of Science, Rehovot, Israel. [2] Institute of Imaging and Computer Vision, RWTH Aachen University, Aachen, Germany. [3] Institute for Automation and Applied Informatics, Karlsruhe Institute of Technology, Karlsruhe, Germany. [4] Department of Life Science Core Facilities, Weizmann Institute of Science, Rehovot, Israel. [5] LIS (UMR 7020), IBDM (UMR 7288), Turing Center For Living Systems, Aix-Marseille University, Marseille, France. ✉email: paul.villoutreix@univ-amu.fr; tstern@princeton.edu; Eli.Zelzer@weizmann.ac.il

Cell morphology and organization plays a major role in tissue and organ morphogenesis[1]. For example, during early fly development, cell rearrangements such as intercalation and rosette formation are the main driving force of germ-band extension, while apical cell constriction drives mesoderm invagination[2–7]. Other notable studies have uncovered cell packing and topology in the plant leaf[8–12], multi-scale branching in the mammalian kidney[13–16], and axis elongation in the avian embryo[17–22]. Thus far, most of these morphogenetic studies have focused on two-dimensional epithelial sheets or early embryogenesis[23–28]. Studying these cellular behaviors in large three-dimensional (3D) tissues is challenging, as the data require the development of a methodology for time-efficient 3D imaging, registration, segmentation, and quantitative analysis of cells. An additional challenge is investigating these processes in mice, where tissues and organs can contain hundreds of thousands of cells, which forms multiple data scales. Focusing on the tissue-level results in loss of subcellular resolution, while focusing on smaller regions at high-resolution results in loss of their relationship to 3D tissue morphology. Recent advances in tissue clearing combined with light-sheet fluorescence microscopy has allowed for high-resolution mapping of fine tissue architectures in intact 3D samples[29–32]. Yet, despite the advances in whole-organ imaging, there is still a lack of tools with which to accurately extract the 3D morphology of cells and identify morphological patterns in large 3D datasets in a time-efficient manner.

The growth plate is an excellent model for studying 3D cell behavior as a driver of tissue morphogenesis. Growth plates are cartilaginous tissues located at either end of developing bones, whose function in bone elongation depends on the sequential morphogenesis of its chondrocytes[33–37]. The growth plate is divided into four zones from its ends towards the center of the bone. Most extreme is the resting zone (RZ), then the proliferative zone (PZ) followed by prehypertrophic (PHZ) and hypertrophic zones (HZ). This arrangement reflects a series of differentiation states that are marked by unique changes in extracellular matrix (ECM) properties and gene expression profiles[33,34,38–41]. Another hallmark of chondrocyte differentiation is a sequence of morphological changes, from small and round in the resting zone, through flat and elongated in the proliferative zone, to round and large in the hypertrophic zone[42–45]. The significance of this morphogenetic process was demonstrated for example in mice lacking key components of the planar cell polarity (PCP) pathway or beta1 integrins. In the proliferative zone of these mutants, chondrocytes fail to flatten, arrange themselves into columns, and no longer orient their short axis parallel to the P-D axis, leading to short and misshapen bones[44–50]. These works demonstrate that chondrocyte shape and orientation are essential drivers of growth plate structure and function.

Several measurements related to hypertrophic cell size were shown to predict longitudinal bone growth, thereby linking chondrocyte morphology to bone morphology. These measurements include hypertrophic cell volume[35,39,51–57], hypertrophic cell diameter along the P-D axis[58–60], and the proliferation rate per column multiplied by the height of the terminal hypertrophic chondrocytes[58–60].

Notwithstanding these important discoveries, comprehensive understanding of growth strategies of differentiating chondrocytes and the relation between their morphology and growth plate structure and function is still lacking. One reason for that is that due to high density of cells and ECM, imaging the growth plate in its entirety has been a major challenge. Additionally, there is a lack of segmentation tools to accurately extract entire cells for analysis.

In this work, we overcome these obstacles by developing a modular imaging and analysis pipeline called 3D MAPs, which enables to explore large datasets of hundreds of thousands of cells, accurately characterizing their morphology while preserving their tissue-level position. Using 3D MAPs, we discovered new cellular behaviors and patterns that contribute to growth plate structure and activity. Moreover, utilizing this pipeline we identified abnormal cellular behaviors of chondrocytes in growth plates of *Gdf5*-null mice, exposing a new role for this gene in skeletogenesis and concomitantly underscoring the ability of 3D MAPs to identify new molecular components in growth plate biology.

## Results

**3D MAPs: morphometric analysis to test for phenotypic significance.** There are likely key aspects of cell morphology that are important for growth plate function, but there is currently no tool suitable for mining this type of data. To that end, we developed a pipeline called 3D morphometric analysis to test for phenotypic significance (3D MAPs). The major steps of 3D MAPs are clearing, imaging, segmentation, and 3D morphometric analysis. To overcome the challenge of imaging a large and dense tissue, we performed tissue clearing, which enabled us to image the entire growth plate by light-sheet microscopy (Fig. 1A and Bi, ii). To accurately segment entire cells, we designed a new segmentation pipeline in an XML pipeline wrapper for the Insight Toolkit (XPIWIT)[61], which automatically segments cells and their nuclei (Fig. 1Biii). Finally, to correlate between cell shape and its position in the growth plate, we extracted various morphological features (Supplementary Table 1) and then projected them onto the cell location in the growth plate, generating 3D morphology maps (Fig. 1Biv). We could then visually identify potentially important patterns in morphology and analyze these features quantitatively and comparatively across space.

**3D MAPs can identify morphogenetic behaviors in the growth plate.** Having segmented successfully cells from each growth plate, we proceeded to characterize various features of cell morphology (listed in Supplementary Table 1) in three growth plates from forelimb and hindlimb bones, namely proximal tibia (PT), distal tibia (DT), and distal ulna (DU). First, we analyzed well-known properties of cells, such as volume, surface area and density. In agreement with previous reports on chondrocyte hypertrophy[39], cells increased their volume on average ninefold from the resting to hypertrophic zone (Fig. 2A). Analysis of the entire growth plate revealed that on average, cells in the proliferative zone grew 20% larger than resting zone cells, prehypertrophic cells grew 74% larger than proliferative zone cells and hypertrophic cells grew 50% larger than prehypertrophic cells. Similarly, we found that cells increased their surface area fivefold on average as they underwent differentiation, increasing by 20% from the resting to proliferative zone, 66% from the proliferative to prehypertrophic zone, and 36% from the prehypertrophic to hypertrophic zone. These surprising results show that contrary to the paradigm that most of the growth occurs in the hypertrophic zone, 65% of chondrocyte total volume enlargement and 70% of total surface area increase occur at earlier stages. Cell density was more variable between the three types of growth plates, but it decreased on average fivefold during differentiation (12-fold at most). In the distal tibia, for example, cell density increased by 7% from the resting to proliferative zone, and then decreased by 60% from the proliferative to prehypertrophic zone, and again by 37% from the prehypertrophic to hypertrophic zone (Fig. 2A). These results show negative correlation between cell volume/surface area and density, likely to preserve bone circumference.

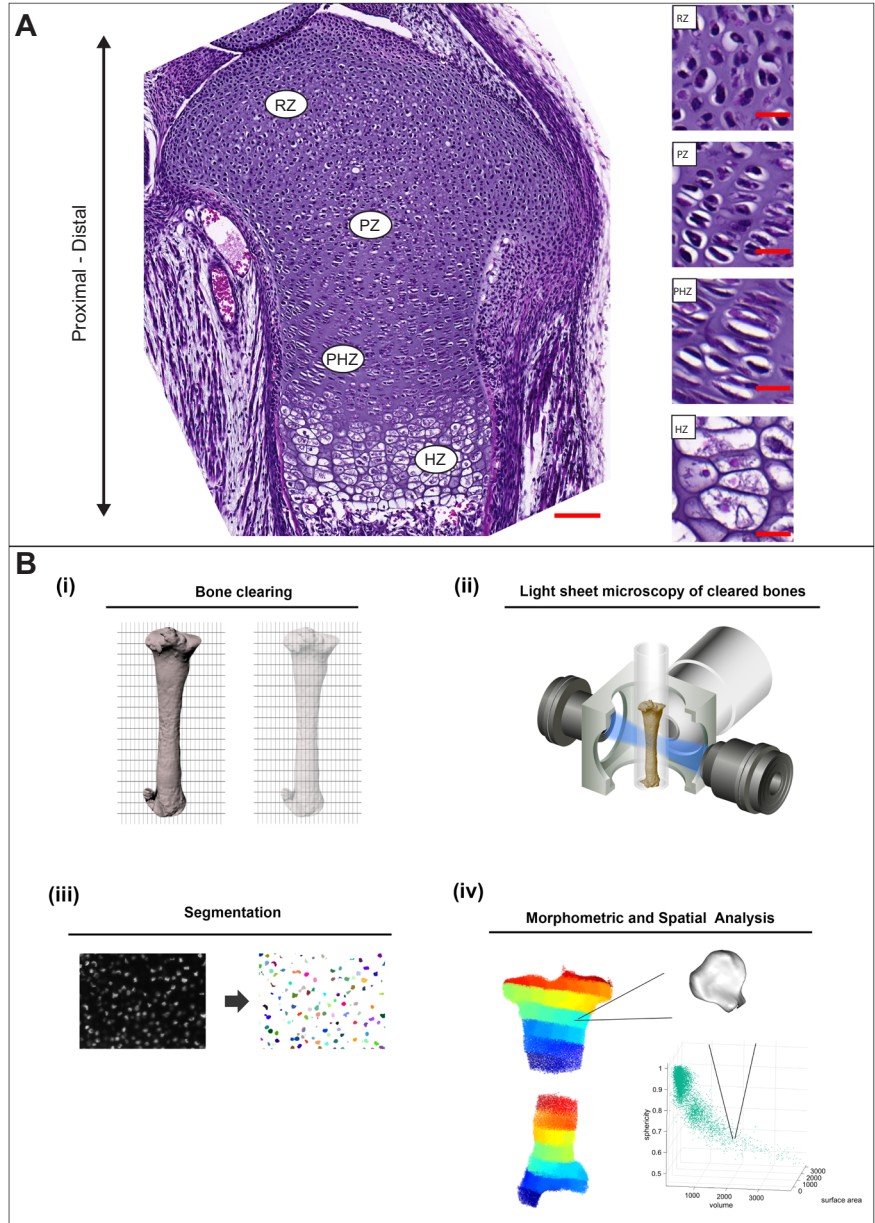

**Fig. 1 3D MAPs: morphometric analysis to test for phenotypic significance. A** 7 μm-thick paraffin sections of an E18.5 proximal tibial growth plate stained with H&E show the typical four zones of the growth plate along the proximal-distal axis of the bone. Scale bar: 100 μm. The magnified images on the right show the morphological changes of chondrocytes during their differentiation sequence, including flattening and column formation of proliferative zone (PZ) chondrocytes, enlargement of column cells in the prehypertrophic zone (PHZ) and increase in size of terminally differentiated chondrocytes in the hypertrophic zone (HZ). Scale bars: 20 μm. **B** An overview of 3D MAPs pipeline. (i) The bone is dissected and cleared by PACT-deCAL, and nuclei are fluorescently labeled. (ii) The cleared bone is embedded in a glass capillary and proximal and distal growth plates are imaged at high-resolution with a Z.1 light-sheet microscope. (iii) Each image stack undergoes nuclear and cellular segmentation. (iv) The segmented cells and nuclei are subjected to morphometric analyses and are projected back onto their anatomical position in the bone.

Hypertrophic cell volume and height are well-established predictors of long bone elongation potential[35,39,51,58–60,62,63]. To compare between the predictive values of these two measurements, we calculated the growth of the PT, DT, and DU growth plates from E16.5 to postnatal day (P) 40 using previously published data[64] (Fig. 2B) and correlated them to hypertrophic cell volumes and heights (i.e., length in the P-D dimension) at E16.5 (Fig. 2C and Supplementary Fig. 4A). For both measurements, we observed similar trends of increasing in the PHZ and peaking in the HZ. However, the volume of hypertrophic cells was found to be correlated with all the observed differences in bone growth (Fig. 2B and Supplementary Fig. 3A, C), whereas cell

diameter was correlated only with some of these differences (Supplementary Fig. 4A). Altogether, these results confirm the ability of 3D MAPs to identify morphogenetic behaviors of cells in the growth plate and show that hypertrophic cell volume is a better predictor of longitudinal bone growth than hypertrophic cell height.

**Allometric and isometric growth behaviors define chondrocyte shape throughout differentiation.** As bone elongation is the output of sequential morphogenesis of chondrocytes, to understand this process it is necessary to characterize the

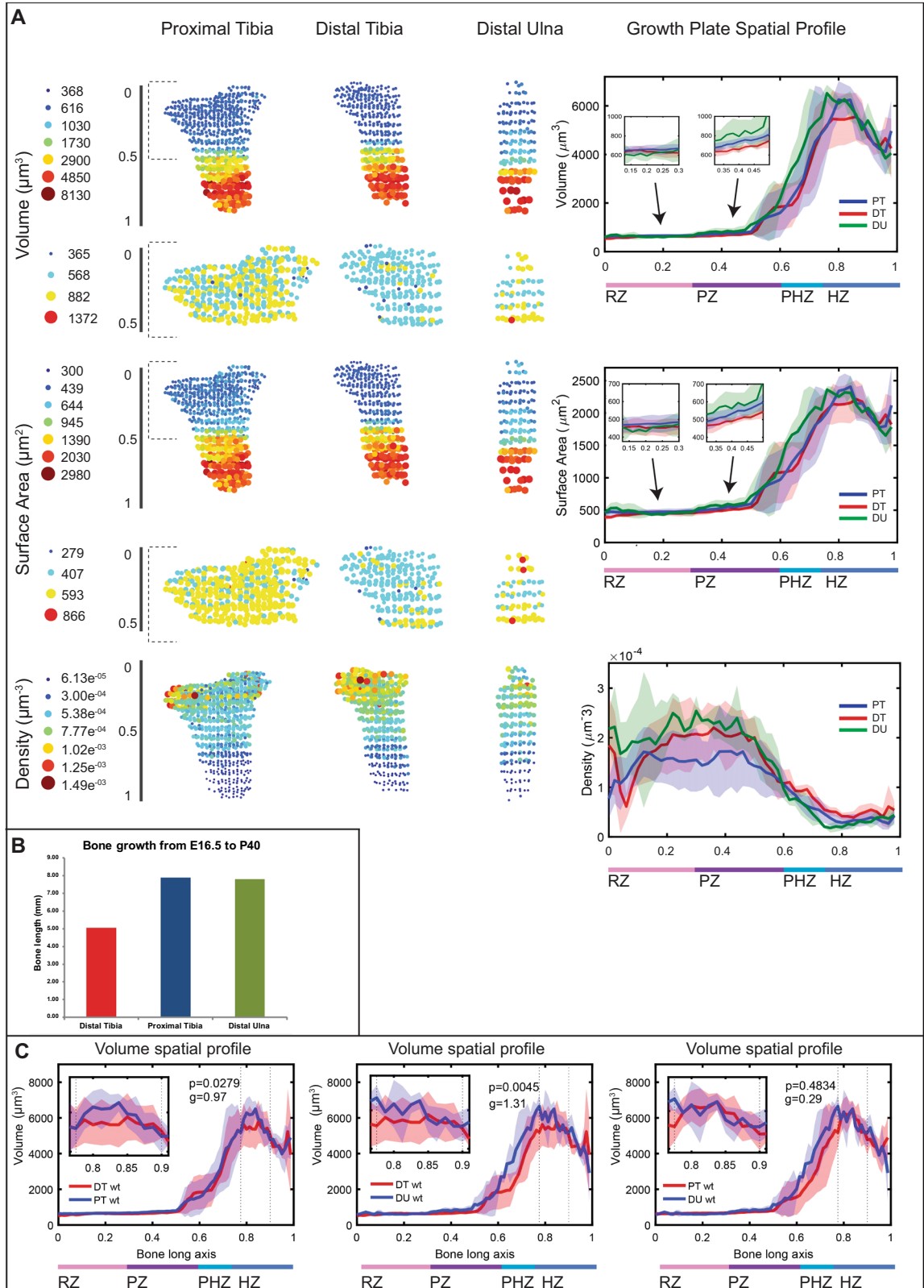

morphological landscape during chondrocyte differentiation. For this, we performed a detailed analysis of chondrocyte growth and morphogenesis.

Histological studies have firmly established that during chondrocyte differentiation, cells change their morphology from round in the resting zone to flattened in the proliferative zone and become more round again in the hypertrophic zone, yet how differentiating chondrocytes transition between morphologies is still unknown[40,42,43,45,65]. Chondrocyte morphogenesis can utilize different growth mechanisms to achieve the same morphology implicating different cellular machineries, and thus understanding these transitions is fundamental. One way to change cell morphology is by isometric growth, where only cell size changes. The other way is allometric growth, where both size and shape

**Fig. 2 3D MAPs can identify morphogenetic behaviors in the growth plate.** Morphology and spatial analysis were performed for each segmented cell and each feature was projected back to the anatomical position of that cell, creating a 3D growth plate map. Each circle represents the mean value within a 75 μm cube. All 3D maps are oriented such that the RZ is at the top and the HZ is at the bottom. Additionally, all growth plates were registered to one another, and a spatial profile was created showing how each feature changes along the differentiation axis of each growth plate. (**A**) Cell volume, surface area, and density were computed for the growth plates of the proximal tibia (PT), distal tibia (DT), and distal ulna (DU). Dashed gray brackets demarcate zoomed-in area of the first half (0–0.5) of each growth plate, highlighting variations in volume and surface area among the different growth plates. Spatial profiles, which were plotted for each feature, show that in all three growth plates, cell volume and surface area increased during chondrocyte differentiation, while cell density decreased. Shaded region shows standard deviation between samples (PT and DT, $n = 5$; DU, $n = 3$). Insets in graphs show zoomed-in regions from the RZ and PZ to highlight the volume and surface area increase. **B** Bar graph shows mean contribution of PT (blue), DT (red), and DU (green) growth plates to total bone growth from E16.5 to P40 ($n = 3$). The PT and DU growth plates were more active than the DT growth plate. **C** The largest 10% of HZ cell volumes (region between dashed lines) were compared by two-tailed Student's $t$-test between means of growth plates. Correlating with the growth plate activity, HZ cell volumes were significantly different between DT and PT ($p = 0.0279$, $g = 0.97$) and between DT and DU growth plates ($p = 0.0045$, $g = 1.31$), but not significantly different between PT and DU growth plates ($p = 0.4834$, $g = 0.29$). Shaded region shows standard deviation from the mean between samples (PT and DT, $n = 5$; DU, $n = 3$).

change. Mathematically, isometric growth occurs when the surface area scales to the two-third power compared to the volume[66–69], while in allometric growth, this scaling factor differs. To determine which growth strategy is used by differentiating chondrocytes, we plotted the mean volume$^{2/3}$ to surface area (Vol$^{2/3}$/SA) ratio along the P-D axis of the growth plate and used linear regression to characterize the growth as isometric (slope = 0) or allometric (slope ≠ 0)[68]. We found that cells grow by volume-dependent allometric growth in the resting and prehypertrophic zones, surface area-dependent allometric growth in the proliferative zone, and isometric growth in the hypertrophic zone (Fig. 3A, B). The distal tibia differed only in the resting zone, where cells grew by surface area-dependent allometric growth. These results show that chondrocytes switch between three main strategies during their differentiation to allow morphological changes in the different zones. The consistency of the results across growth plates suggests the generality of this finding, whereas the difference in the distal tibia resting zone may reflect its different growth rate, as compared to the two other growth plates.

To reveal the specific mechanism by which allometric growth changes cell morphology, we extracted the three cell axes and followed changes in the ratios between them during allometric growth. For that, we conducted principal component analysis (PCA) on the masked region of the segmented cell (Fig. 3C), where PC3 described the short cell axis, PC2 described the medium axis and PC1 described the long axis. This analysis identified large variances of at least twofold between all three principal components, suggesting that their assignments could be used to represent the three cell axes (Fig. 3C). To capture morphological changes during allometric growth, the PC coefficients were plotted as ratios and compared along the growth plate (PC3/PC1, PC2/PC1, and PC3/PC2) (Fig. 3D, E and Supplementary Fig. 4B). Results showed that cells in the proliferative zone flattened by shortening their short axis while stretching their long and medium axes at the same rate. In contrast, cells in the resting and prehypertrophic zone changed their morphology by swelling along their short and medium axes faster than along their long axis. This occurs faster in the prehypertrophic zone. Altogether, we found that during differentiation, chondrocytes change their morphology continuously and dramatically by employing multiple growth strategies.

**Dorsal-ventral polarization of long cell axis in the growth plate.** Cell orientation in the growth plate is important, especially in the proliferative zone, where cells orient their shortest axis parallel to the long axis of the bone to form columns, thereby facilitating bone elongation[50,70–72]. Previous studies also showed that the cell division occurs perpendicular to the axis of growth. However, these analyses were performed in 2D, so nothing is known about the orientation of the third cell axis and how this

relates to what was previously described. To study in 3D the orientations of differentiating chondrocytes, we registered the growth plates and quantified the orientation of the three orthogonal axes of each cell using PCA (Fig. 4A).

To verify that the imaging angle does not introduce an artifact, we analyzed an ulna imaged at two orthogonal rotations. Results showed that the orientation of PC1 was not changed between the two imaging angles, which was further validated quantitatively by performing a segmentation error analysis (Supplementary Figs. 2g and 5), excluding the possibility of an artifact.

To identify deviations in orientation, color maps were normalized to represent the angle between a cell's vector and the mean vector of all cells in the growth plate. In agreement with previous studies[44,50,72], we found that proliferative zone cells align their short axis (PC3) along the proximal-distal axis of the bone (Fig. 4B) with very little variation (indicated by the thin lines). In the resting and hypertrophic zones, we observed high variability in short axis orientation (indicated by thick lines) and overall disorder. Prior works showed that the direction of cell intercalation in the proliferative zone is along the medial-lateral axis[44,48,73]. Interestingly, we found that this plane represented the medium axis of the cell (PC2). The orientation of this axis in the proliferative zone was highly uniform along the medial-lateral bone axis, whereas like the short axis, in the resting and hypertrophic zones it was disordered (Fig. 4B).

Strikingly, we found that throughout the growth plate, the cells oriented their long axis (PC1) towards the dorsal-ventral bone axis, with some variability occurring in the resting and hypertrophic zones (Fig. 4B). Notably, the orientation patterns of all three axes were conserved in the three analyzed growth plates. Altogether, by analyzing 3D cell orientation we discovered a new cell behavior in which the long cell axis points towards the dorsal-ventral axis in all zones and in different growth plates, suggesting a biological need for this behavior.

**Gdf5 expression is necessary for normal SMAD 1/5/9 activity in the growth plate.** To identify new regulators of cell growth and morphology and to demonstrate the sensitivity of 3D MAPs to detect abnormalities in these processes, we sought to study growth plates of mutants for genes whose loss of function leads to abnormal skeletal development, but without known involvement in growth plate biology. A search of the literature revealed that loss of *Gdf5*, a well-known regulator of joint formation[74–78], also causes shortening of tibias[79], although histologically the growth plates of these mice appeared unaffected[80]. *Gdf5* is known to be expressed only in the joint and outer collar region[74,75,77–79,81]. To assess the possible effect of GDF5 on the growth plate, we first revisited the question of *Gdf5* expression in this tissue. For that, we used the hybridization chain reaction (HCR) method[82] to

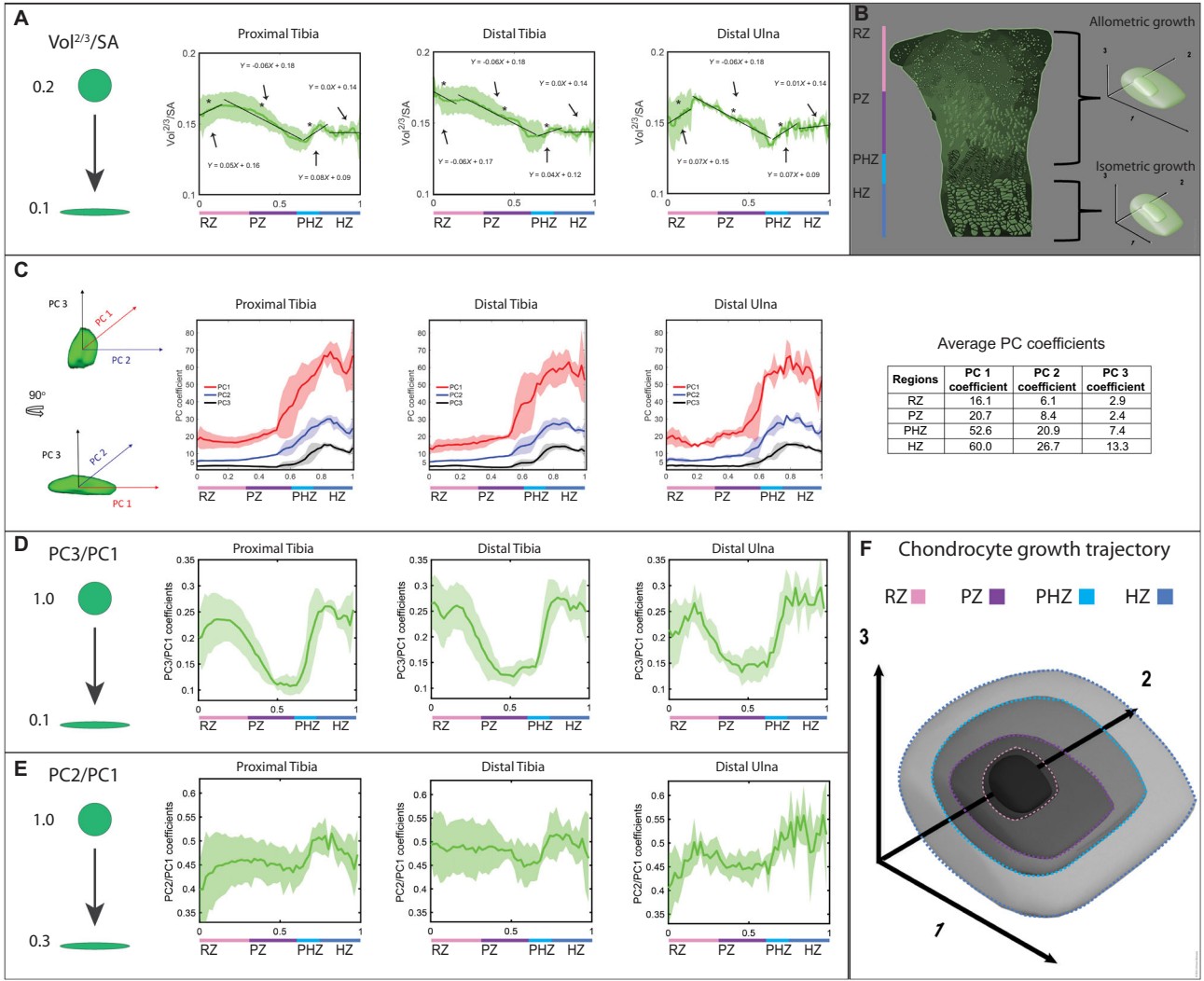

**Fig. 3 Allometric and isometric growth behaviors define chondrocyte shape throughout differentiation. A** The Vol$^{2/3}$/SA ratio was plotted as a spatial profile along the differentiation axis and regression lines were fit for each change in growth behavior to determine if the growth was isometric (slope = 0) or allometric (slope ≠ 0). Slopes > 0 represent volume-dependent allometric growth, whereas slopes < 0 represent surface area-dependent allometric growth. In the PT and DU, RZ cells grew by volume-dependent allometric growth (PT slope = 0.05 $p$ = 5.92e$^{-09}$ $R^2$ = 0.93; DU slope = 0.07 $p$ = 7.37e$^{-04}$ $R^2$ = 0.6), while in the DT, RZ cells grew by surface area-dependent allometric growth (DT slope = –0.06 $p$ = 1.06e$^{-02}$ $R^2$ = 0.41). In the PZ, cells grew by surface area-dependent allometric growth (PT slope = –0.06 $p$ = 2.89e$^{-31}$ $R^2$ = 0.95; DT slope = –0.06 $p$ = 5.71e$^{-33}$ $R^2$ = 0.95; DU slope = –0.06 $p$ = 2.06e$^{-34}$ $R^2$ = 0.96), in the PHZ by volume-dependent allometric growth (PT slope = 0.08 $p$ = 8.09e$^{-05}$ $R^2$ = 0.68; DT slope = 0.04 $p$ = 3.11e$^{-04}$ $R^2$ = 0.62; DU slope = 0.07 $p$ = 8.64e$^{-04}$ $R^2$ = 0.56), and in the HZ cells grew by isometric growth (PT slope = 0 $p$ = 0.937 $R^2$ < 0.01; DT slope = 0 $p$ = 0.903 $R^2$ < 0.01; DU slope = 0.01 $p$ = 0.301 $R^2$ = 0.06). **B** Scheme illustrating that chondrocytes change their shape as they grow allometrically in the RZ, PZ, and PHZ, but maintain their shape while growing isometrically in the HZ. **C, D** Principal component analysis (PCA). **C** PC1 (red arrow) of the object represents the long cell axis, PC2 (blue arrow) the medium axis, and PC3 (black arrow) the short axis. Spatial profiles of PC 1, 2, and 3 show that they are significantly larger than one another and they are conserved among the three growth plates. The table shows the mean PC coefficients across all three growth plates per zone. **D** In all three growth plates, as cells differentiated from the RZ to the PZ, they decreased their PC3/PC1 ratio by half and then returned to the same ratio when they differentiated to the PHZ and HZ. **E** In the PT and DU, cells increased their PC2/PC1 ratio in the RZ and slightly decreased their ratio in the DT RZ. In all growth plates, the PC2/PC1 ratio was constant throughout the PZ, and increased as cells differentiate to the PHZ. The ratio at the end of the HZ is the same as in the RZ. **F** Scheme illustrating how each cell axis changes during growth and differentiation in the growth plate. Shaded region shows standard deviation from the mean between samples (PT and DT, $n$ = 5; DU, $n$ = 3). $p$-values were calculated by two-tailed Student's $t$-test between mean slopes and slope = 0. Asterisks denote statistical significance. Abbreviations: PT, proximal tibia; DT, distal tibia; DU, distal ulna; RZ, resting zone; PZ, proliferative zone; PHZ, prehypertrophic zone; HZ, hypertrophic zone. Illustration by Chinami Michaels, © 2020 Chinami Michaels Biomedical Visualization Studio, [ChinamiMichaels.com](ChinamiMichaels.com), provided under CC BY 4.0.

measure single-molecule fluorescent in situ hybridization of *Gdf5* in E16.5 tibias. Surprisingly, we found that *Gdf5* is expressed in all growth plate zones (Fig. 5A–H and Supplementary Fig. 6A). Quantification showed high variability in expression levels, from up to 100 transcripts per cell in the outer collar to 0-2 transcripts per cell in the center of the growth plate (Fig. 5H).

To understand the significance of this newly discovered *Gdf5* expression in the growth plate, we compared the activity of its downstream effectors SMAD 1/5/9 between control and *Gdf5* knockout (KO) growth plates (Fig. 5I and Supplementary Fig. 6B). Interestingly, we identified reduced phosphorylation of SMADs throughout the *Gdf5* KO growth plate, which supported the

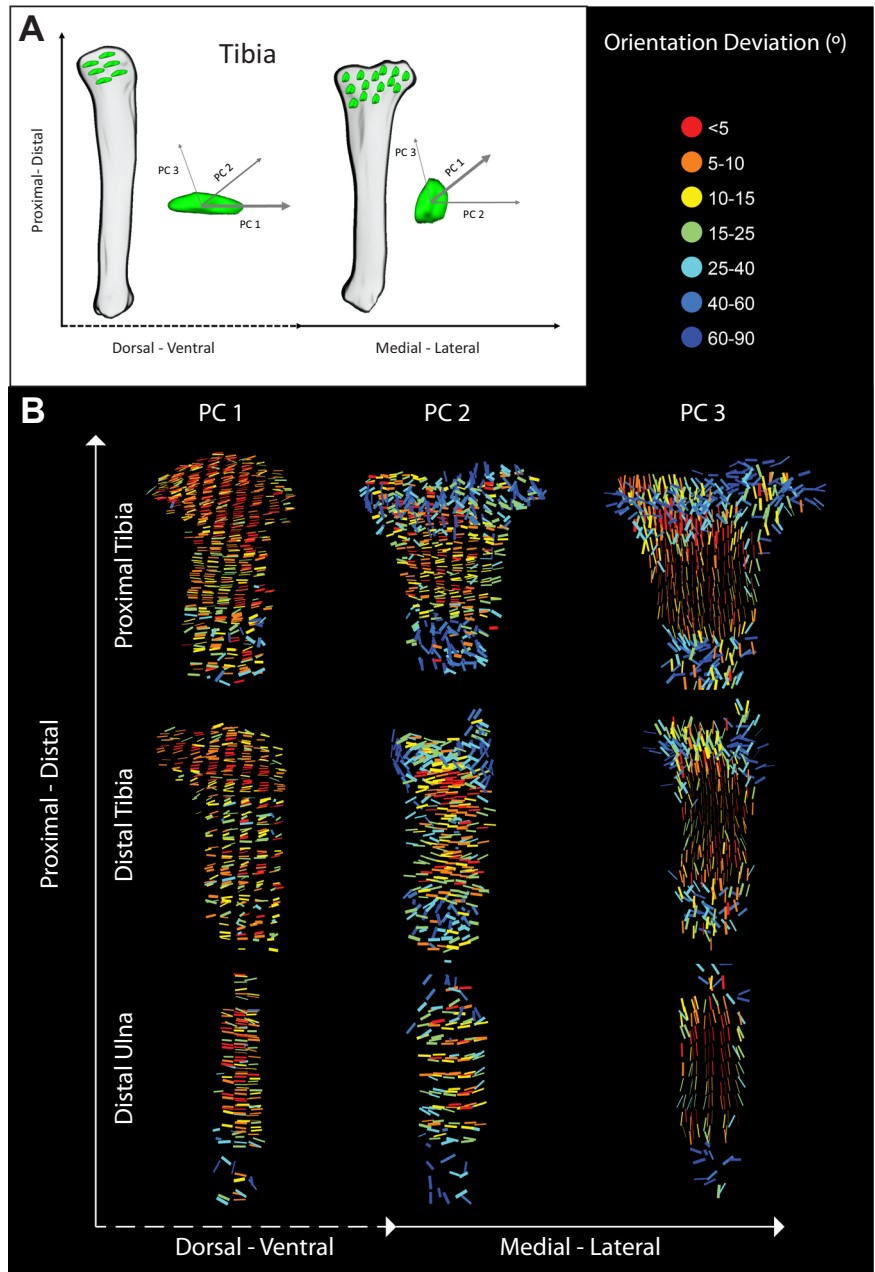

**Fig. 4 Dorsal-ventral polarization of long cell axis in the growth plate.** Cell orientation was calculated by extracting the three orthogonal axes of each cell from the segmented object using principle component analysis (PCA). PC1 (thickest arrow) of the object represents the long axis, PC2 (thinner arrow) the medium axis, and PC3 (thinnest arrow) the short axis. The schematic drawing of the tibia in **A** shows that bone alignment determines which orientation axes can be viewed in 2D. **B** Cell orientation along the three axes (PC 1, 2, and 3) was plotted for all three growth plates (proximal tibia, distal tibia, and distal ulna). In all cases, the growth plates were placed such that the RZ pointed upward and the HZ pointed downward. The colormap is normalized to each sample, representing the deviation in degrees from the average orientation of each growth plate. All three growth plates exhibited similar cell orientation behaviors. The long cell axis was aligned to the dorsal-ventral axis of the bone. The medium axis was highly variable in the RZ and HZ but in the PZ, it was aligned uniformly to the medial-lateral axis of the bone. The short axis was also highly variable in the RZ and HZ but in the PZ, it was aligned uniformly towards the proximal-distal bone axis. RZ, resting zone; PZ, proliferative zone; PHZ, prehypertrophic zone; HZ, hypertrophic zone.

possibility that GDF5 is involved in regulating chondrocyte differentiation and morphogenesis. The loss of SMAD activation was much stronger than what would be expected based on the *Gdf5* expression pattern within the growth plate. Thus, it is reasonable to assume that loss of *Gdf5* leads to reduction in other BMP family members that are expressed in the growth plate[83,84].

**Gdf5 regulates chondrocyte growth and morphology during differentiation.** To study the possibility that these molecular changes translate to morphological changes, we applied 3D MAPs to E16.5 *Gdf5* KO embryos[74], which already exhibited tibial deficits (Supplementary Fig. 7A, B). We established a database of hundreds of thousands of mutant cells, which we compared to the control database. Comparing proximal and distal growth plates of *Gdf5* KO and control tibias revealed several significantly different features. Although mean cell volume in the resting and

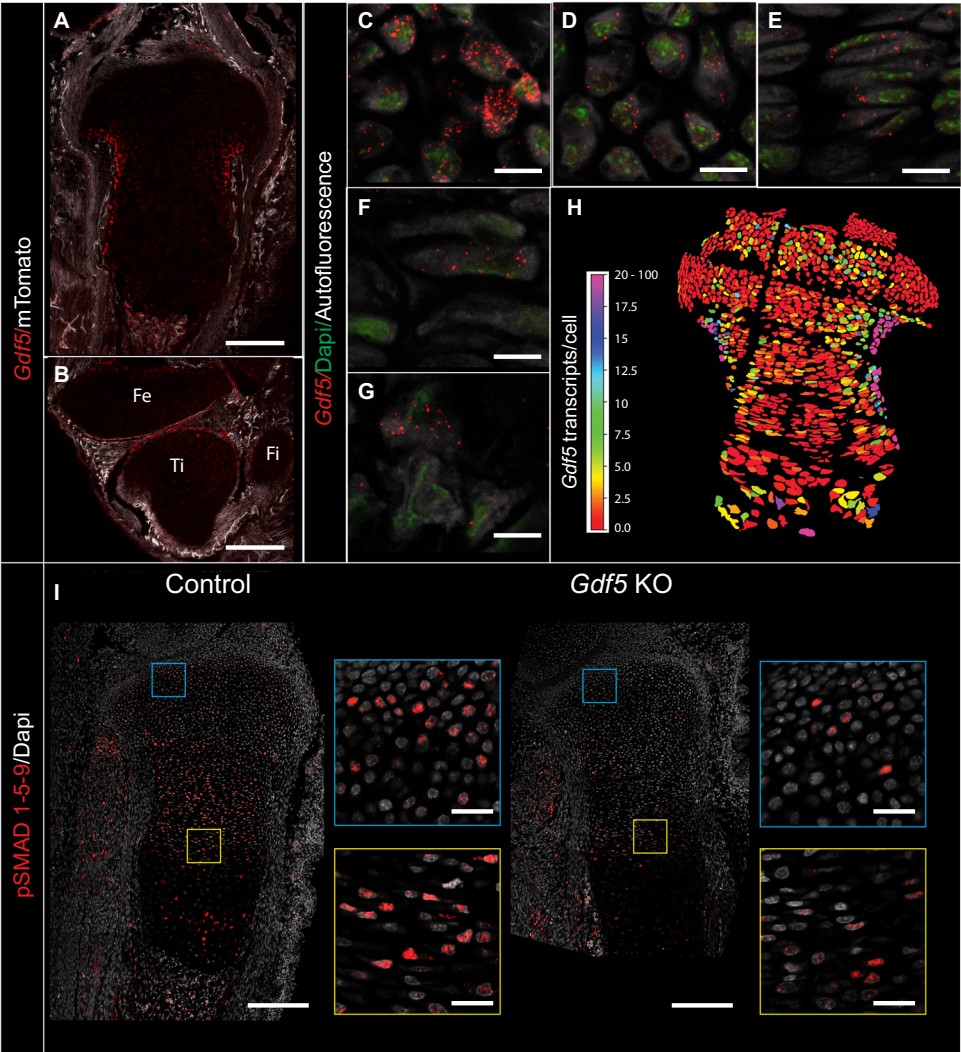

**Fig. 5 GDF5 is expressed in the growth plate and regulates SMAD 1/5/9 activity. A**, **B** HCR smFISH for *Gdf5* in the proximal tibia (**A**) and knee joint (Fe, femur; Ti, tibia; Fi, fibula) (**B**). Scale bars: 200 μm. **C**–**G** *Gdf5* is expressed in the collar region (**C**), resting zone (**D**), proliferative zone (**E**), prehypertrophic zone (**F**), and hypertrophic zone (**G**) (*n* = 4 biologically independent samples examined over 3 independent experiments). Scale bars: 10 μm. **H** A heat map shows the number of *Gdf5* transcripts per cell throughout the proximal tibia growth plate (*n* = 3). The highest signal per cell is in the collar region, followed by the proliferative zone adjacent to the collar, and some areas in the resting zone. Many cells in the center have between 0 and 2 transcripts. **I** Immunofluorescence for pSMAD 1/5/9 shows a reduction in the proximal tibia of *Gdf5* KO mice compared to controls (*n* = 4 biologically independent samples examined over three independent experiments). Scale bar: 200 μm. Blue and yellow square insets show zoomed-in regions in the resting and proliferative zones, respectively. Scale bar: 25 μm.

proliferative zones was comparable, the standard deviation was double that of the mutant in the direction of larger volumes (Fig. 6A). Conversely, control growth plates had significantly larger volume standard deviations in the prehypertrophic zone compared to mutants, indicating that the capacity of cells to increase their volume was higher in controls. Finally, the hypertrophic zone mutant cells were on average 30% smaller in the proximal tibia relative to the control (Fig. 6A). Since hypertrophic cell volume is a known determinant of growth plate height, these discrepancies can explain why the tibias are shorter in the absence of *Gdf5*. The same trends were observed for cell surface area (Supplementary Fig. 7C).

Since the mean cell volume was similar in control and mutant resting and proliferative zones, we did not expect to find major differences in cell density. To our surprise, cell density was significantly higher in the resting and proliferative zones of the mutant (Fig. 6B, C). This suggests either a reduction in ECM, or tighter cell packing. Conversely, in the hypertrophic zone of the

same growth plates, mutant cells had smaller cell volumes and were less dense compared to controls, likely contributing to the shortening of the mutant tibias.

As *Gdf5* KO tibias are shorter than controls and have abnormal cell volume, surface area, and density, we next investigated if the cell growth mechanisms were impaired. Comparing the vol$^{2/3}$/SA ratios between controls and mutants revealed that although hypertrophic cells were smaller, they grew normally by isometric growth, suggesting that defects at earlier stages are responsible for the aberrant growth. In the proliferative zone, proximal and distal tibias grew allometrically by surface area increase as in controls, but their growth rate was significantly reduced, resulting in a lesser degree of flattening (Fig. 6E). This was supported by analyzing cell sphericity, which showed that mutant cells in the resting and proliferative zones were on average 8% more spherical than control cells (Fig. 6D and Supplementary Fig. 7D, 3D'). The two growth plates differed in the resting and prehypertrophic zones. While cells in the proximal tibia resting zone grew

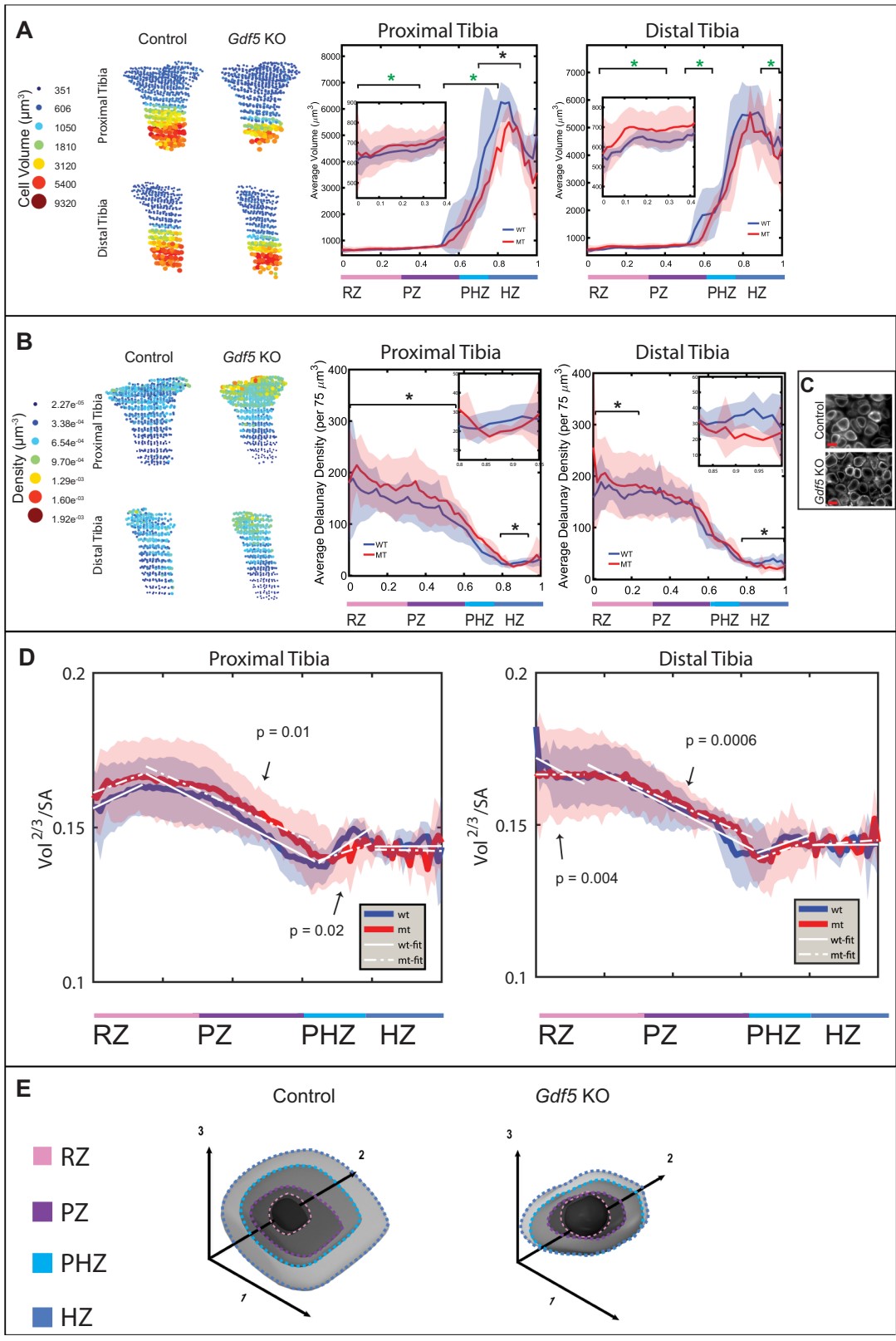

normally, cells in the distal tibia grew isometrically instead of allometrically (Fig. 6D), resulting in cell growth without a change in shape. On the other hand, while cells in the distal tibia prehypertrophic zone grew normally, cells in the proximal tibia grew at half the rate of controls by volume-dependent allometric growth. This severe growth defect explains why hypertrophic cell volume was significantly reduced in the proximal tibia. Altogether, these findings show that GDF5 plays a role in regulating both surface area- and volume-dependent allometric growth mechanisms, and that growth defects at early stages of chondrocyte differentiation have a cumulative and severe effect on growth plate activity.

**Fig. 6 *Gdf5* regulates chondrocyte growth and morphology during differentiation.** Representative 3D maps and comparative spatial profiles of control and mutant samples show that *Gdf5* KO tibias display abnormal cell volume (**A**), density (**B** and **C**), and growth mechanisms (**D**). In the proximal (PT) and distal (DT) *Gdf5* KO growth plates (**A**), the standard deviation of cell volume was abnormally high in the RZ (PT $p = 4.8e^{-06}$; DT $p = 3.7e^{-12}$) and DT HZ ($p = 0.016$), while it was low in the PT PHZ ($p = 2.2e^{-04}$) and DT PZ ($p = 0.014$). The mean cell volume was abnormally low in the PT HZ ($p = 0.038$). **B** 3D maps and spatial profiles show that the RZ and PZ of *Gdf5* KO growth plates display significantly higher cell density compared to controls (PT $p = 6.0e^{-05}$; DT $p = 0.001$). Additionally, the HZ has lower cell density (PT $p = 0.034$; DT $p = 5.5e^{-05}$). **C** Light-sheet images of cells of control and *Gdf5* KO growth plates highlights the higher density in the mutant RZ. Scale bar, 10 μm. Black asterisks denote significant *p*-values calculated by two-tailed Student's *t*-test between means and green asterisks denotes significant *p*-values calculated by two-tailed Student's *t*-test between standard deviations (control, $n = 5$; mutant, $n = 4$). **D** Spatial profile of vol$^{2/3}$/SA ratio with regression line fitting for each zone shows that in *Gdf5* KO growth plates, cells grew by volume-dependent allometric growth in the PT RZ (slope = 0.04 $p = 7.43e^{-06}$ $R^2 = 0.8$), by isometric growth in the DT RZ (slope = 0.0 $p = 0.9$ $R^2 < 0.01$), by surface area-dependent allometric growth in the PT and DT PZ (PT slope = −0.05 $p = 4.75e^{-29}$ $R^2 = 0.93$; DT slope = −0.05 $p = 1.89e^{-31}$ $R^2 = 0.95$), by volume-dependent allometric growth in the PT and DT PHZ (PT slope = 0.04 $p = 6.44e^{-04}$ $R^2 = 0.58$; DT slope = 0.04 $p = 2.81e^{-05}$ $R^2 = 0.73$), and by isometric growth in the PT and DT HZ (PT slope = 0.0 $p = 0.822$ $R^2 < 0.01$; DT slope = 0.01 $p = 0.422$ $R^2 = 0.04$). The mutant growth behavior was significantly different from controls in the DT RZ ($p = 4.0e^{-03}$), PT and DT PZ (PT $p = 0.01$; DT $p = 6.0e^{-04}$), and PT PHZ ($p = 0.02$). **E** Scheme illustrating how each cell axis changes during growth and differentiation in control and *Gdf5* KO growth plates. *p*-values were calculated by two-tailed Student's *t*-test between mean slopes (control, $n = 5$; mutant, $n = 4$). Shaded regions denote standard deviation from the mean. WT, wild type; MT, mutant. Illustration by Chinami Michaels, © 2020 Chinami Michaels Biomedical Visualization Studio, [ChinamiMichaels.com](ChinamiMichaels.com), provided under CC BY 4.0.

**GDF5 regulates chondrocyte polarity in the growth plate**. Since cell growth and flattening are tightly related to cell orientation in the growth plate[44,46–49], we proceeded to examine the latter. Indeed, we found a large variability in cell orientation along all three axes in the different zones of the mutant growth plate, relative to control (Fig. 7A, B). As abnormal orientation could result in aberrations in column structure, we examined cross-section maps from the proliferative zone. As seen in Fig. 7A(i-l) and 7D, whereas in control columns, cells aligned their short axis parallel to the proximal-distal bone axis, mutant cells pointed 25–40 degrees away in some areas. In the other cell axes mutant cells varied up to 40 degrees, as compared to 15 degrees in control cells (Fig. 7A(a–h)). Finally, while in controls the PC1 lines laid one on top of another, indicating that the columns were aligned along the proximal-distal axis, we observed a fan-like pattern in the mutant, supporting the possibility that they formed abnormal columns (Fig. 7A(a–d) and D).

To validate these statistical results, we performed histological analysis on control and mutant growth plates at P6, when the column structure is already well-established (Fig. 7C). Alcian blue staining clearly showed that mutant columns were disorganized compared to controls. Overall, we observed defects in cell orientation and column formation that might impair growth plate function, leading to the observed growth defects in *Gdf5* KO tibias (Fig. 7D).

## Discussion
The tight connection between cell and organ morphogenesis[2,3,5,6] highlights the need to characterize in detail the 3D morphology of cells in the spatial context of the tissue. In this work, we established a new pipeline called 3D MAPs, which enabled us to overcome these limitations and create complex multiresolution datasets of the 3D morphology of growth plate cells throughout their differentiation. By analyzing these datasets, we show that chondrocytes use both allometric and isometric growth strategies during differentiation in a cell axis-specific manner. Additionally, 3D MAPS revealed a striking conservation in the long cell axis orientation across all zones of the growth plate. Finally, we provide an explanation for the shortening of *Gdf5* KO tibias by identifying a reduction in SMAD 1/5/9 activity together with multiple abnormalities in cell growth, shape and organization.

Nearly a century ago, Stump proposed that "The growth impulse affecting the shape, size, and form of the bone is inherent in the cartilage cells, both in diaphysis and epiphysis."[85]. In the last few decades, there have been several attempts to study 3D morphology of cells in the growth plate[42,43,86–88]. However, these efforts focused on comparing small discrete regions from each zone and, therefore, the continuous process of morphogenesis was not observed. Our work fills this gap and provides the full sequence of morphogenesis during chondrocyte differentiation.

Analysis of cell morphogenesis in the entire growth plate revealed that chondrocytes choose between allometric and isometric growth. Since isometric growth was found to be employed only in the hypertrophic zone, chondrocytes undergo changes in shape only in the first three zones of the growth plate. Adding to the complexity, allometric growth is driven either by volume or surface area. These two parameters implicate different cellular mechanisms, the former involving rapid water uptake and the latter involving the cytoskeleton.

The importance of using different growth regimes is shown by the finding that resting and hypertrophic chondrocytes have the same aspect ratios, i.e., the same shape. This suggests that in theory, chondrocytes could have grown isometrically throughout the growth plate. A possible explanation for choosing to grow allometrically is to prevent expansion of the bone circumference during elongation.

Another interesting finding is that 65% of total volume increase occurs before chondrocytes reach the hypertrophic zone. Thus far, most studies of cell volume have focused on hypertrophic chondrocytes. Our findings highlight the need to observe earlier stages of chondrocyte differentiation, especially in the prehypertrophic zone, where 50% of the total growth occurs. This notion is supported by[52], who found that cell volume enlargement in the proliferative zone was a major contributor to bone elongation in embryonic chicks. Another example for the importance of studying cell volume throughout the growth plate is our analysis of *Gdf5* KO growth plates. We show that although the isometric growth of hypertrophic chondrocytes was unaffected, their volume was reduced by 30% because of a 50% growth lag in the prehypertrophic zone.

Analyzing cell axis orientation, we found that in the proliferative zone, the axis that orients along the M-L bone axis, which previous studies in 2D suggested as the cell's major axis[44,48], is only the medium axis. The long cell axis, which is more than twice the length of the medium axis, is oriented towards the D-V bone axis. Surprisingly, we discovered that unlike the two other axes, this orientation was conserved in all growth plate zones. Although the reason for chondrocytes to align their long axis along the D-V bone axis is unknown, as is whether this orientation persists at later ages, we speculate that it may provide a structural advantage by increasing the stability of the growth plate during development. Another question related to this phenotype is the molecular mechanisms that regulate it. The

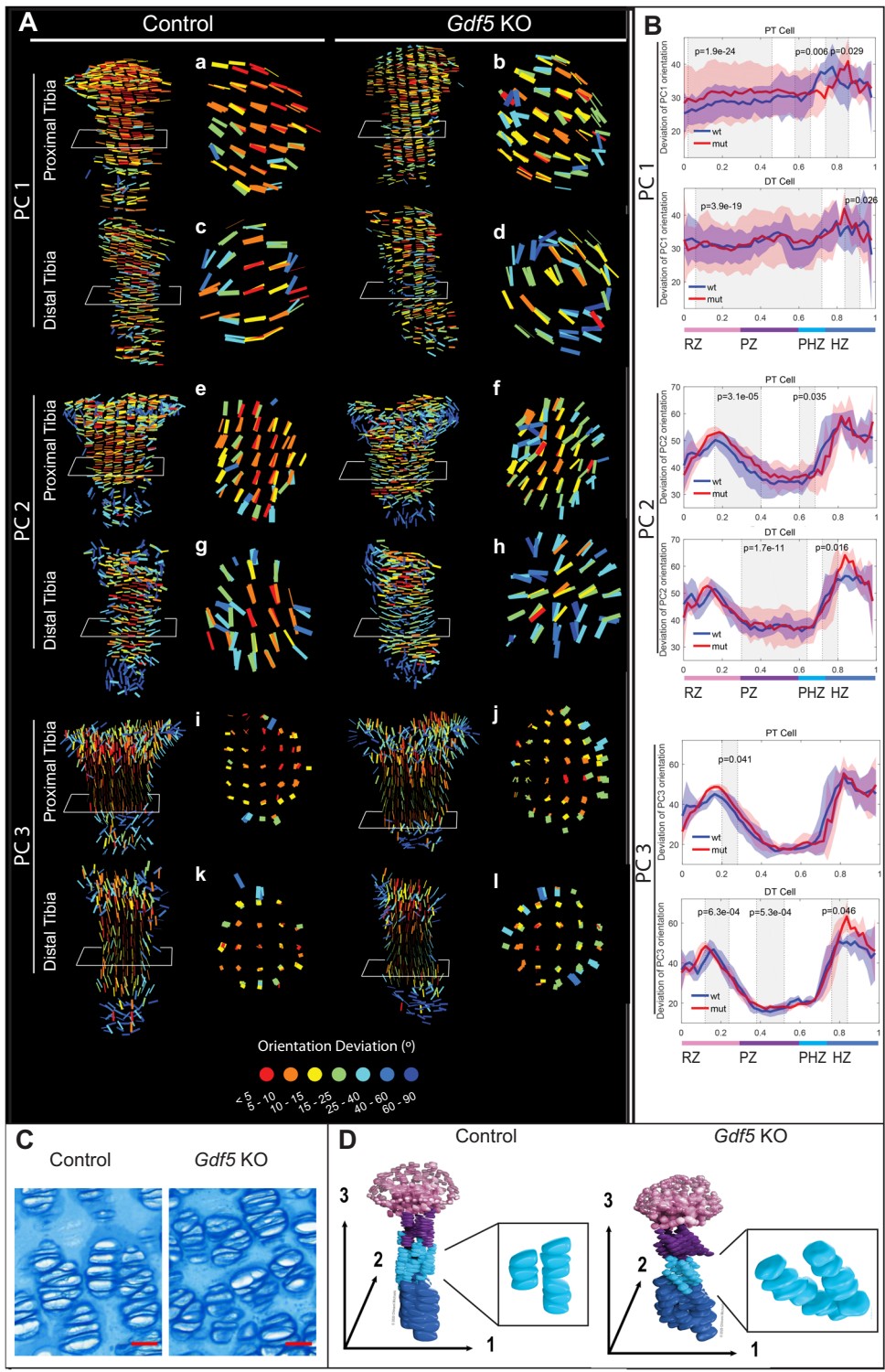

immediate suspects are polarity-inducing molecules, such as members of PCP pathway and integrin molecules, as well as ECM stiffness[73] and mechanical signaling[47], which were shown to be involved in chondrocyte shape and orientation[44,46,48,49].

In contrast to the PCP pathway, GDF5 is not known to regulate chondrocyte orientation. Surprisingly, we found that chondrocytes in the *Gdf5* KO were misoriented along all three axes. This was likely due to impaired cell flattening, leading to abnormal column formation. The mechanism by which GDF5 affects cell orientation is still unknown. Our finding that GDF5 is

expressed by chondrocytes throughout differentiation, together with reduction in SMAD 1/5/9 activity in the *Gdf5* KO growth plates, provides a molecular entry point into this mechanism. The observations that chondrocytes in the *Gdf5* KO growth plates displayed reduced hypertrophic volume and abnormal orientation and column formation provide an explanation for the shortening of tibias in the mutant. Additionally, these results may explain some of the phenotypes observed in Grebe syndrome[75,79–81,89,90].

**Fig. 7 GDF5 regulates chondrocyte polarity in the growth plate. A** Representative 3D maps compare cell orientation of E16.5 PT and DT growth plates between control and *Gdf5* KO mice. Cross sections through the PZ show that cells in the *Gdf5* KO growth plates orient abnormally and with high variation in the long and medium axes (PC1 and PC2) **a**–**l** The growth plates were placed such that the RZ pointed upward and the HZ pointed downward. The colormap is normalized to each sample, representing the deviation in degrees from the mean orientation of each growth plate. Cross sections through *Gdf5* KO growth plates (**b** and **d**) show that long axis (PC1) lines form a fan-like pattern. This, along with the wide range of map colors, indicate that mutant cells lose their polarity along the long axis. A similar behavior was observed for the medium axis orientation PC2, and the effect was larger in the DT (**g** and **h**). Cells in control growth plates aligned their short axis (PC3) along the proximal-distal axis of the bone within a range of 5 degrees (**i** and **k**). Strikingly, cells in *Gdf5* KO growth plates were polarized in this axis, but they deviated from controls by 25–40 degrees (**j** and **l**). **B** Spatial profiles show how all cell axes differ between control and *Gdf5* KO growth plates. The long cell axis orientation (PC1) was significantly more variable in the mutant in multiple locations throughout all zones in both PT and DT growth plates (PT, $p = 1.9e^{-24}$, $p = 0.006$, and $p = 0.029$; DT, $p = 3.9e^{-19}$ and $p = 0.026$). The mutant medium cell axis orientation (PC2) was significantly more variable only in PHZ of the PT ($p = 0.035$) and in the PZ and HZ of the DT ($p = 1.7e^{-11}$ and 0.016). The control medium cell axis orientation was significantly more variable in the RZ of the PT ($p = 3.1e^{-05}$). In the mutant PT, the variability of the short cell axis (PC3) was significantly lower in the RZ and PZ (PT RZ $p = 0.041$; DT RZ $p = 6.3e^{-04}$ PZ $p = 5.3e^{-04}$), but higher in the HZ ($p = 0.046$). **C** Alcian blue staining of the PZ from control and *Gdf5* KO tibias shows abnormal column structure in the mutant. Scale bar, 20 μm. ($n = 2$ biologically independent samples examined over two independent experiments). **D** 3D model of cell orientation in control and *Gdf5* KO growth plates. Insets highlight columns in the proliferative zone. Abbreviations: wt, control; mut, *Gdf5* KO. *p*-values were calculated by two-tailed Student's *t*-test between standard deviations from the mean (shaded regions) of control ($n = 5$) and mutant samples ($n = 4$). Illustration by Chinami Michaels, © 2020 Chinami Michaels Biomedical Visualization Studio, ChinamiMichaels.com, provided under CC BY 4.0.

The 3D MAPs pipeline offers solutions to the multiple problems and challenges in studying 3D cell morphology in the context of the entire tissue. First, it provides a modular tool kit with which to image, register, segment, and quantify morphological properties of hundreds of thousands of cells in a time-efficient manner. Multiresolution capabilities allow the user to shift from single-cell to tissue level to identify morphological patterns amongst thousands of cells within the tissue. Moreover, 3D MAPs allows quantification of 17 different morphological parameters, thereby enabling one to measure the particular contribution of each differentiation step or different morphology parameters to cell growth and tissue morphogenesis. For example, we showed that hypertrophic cell volume was better able to predict longitudinal bone growth than the height of these cells. A third method to predict potential bone growth is the proliferation rate per column multiplied by the height of terminal hypertrophic chondrocytes. However, since this method depends on the ability to identify columns, it should be used only postnatally, once the mature uniclonal columns are established[59,91]. Finally, our method provided a high-resolution map of the morphogenetic sequence that differentiating chondrocytes undergo, allowing the identification of the exact stage and type of cellular abnormality in different mutants. This feature is particularly important as it can point to the molecular mechanism underlying the phenotype.

Regarding the limitations of 3D MAPs, it is a modular pipeline whose resolution and sensitivity depend on imaging resolution and accuracy of segmentation. Using the current imaging and labeling techniques, we were unable to accurately quantify nuclear orientation and PC coefficients. This could be improved by imaging at a higher resolution. Another limitation relates to time-efficient extraction of morphometric parameters from big data, as larger samples with more cells will take longer to clear, image, and analyze. Analysis time could possibly be reduced by running the segmentation and 3D MAPs algorithms on cluster computers. Finally, 3D MAPs alone does not provide detailed molecular information. However, combining labeling of molecular components (e.g., by FISH, immunohistochemistry, or multiple endogenous reporters) with the built-in analysis of morphology of labeled structures of interest offered by 3D MAPs would allow to overcome this obstacle.

In summary, this work introduces a pipeline that allows in-depth profiling of cell growth, morphology, and organization in the spatial context of a mammalian tissue. 3D MAPs provides a means to explore how the growth plate works and to identify the cellular mechanisms that regulate it. As a proof of concept,

analysis of mutant mice identified GDF5 as a regulator of growth plate activity.

## Methods

**Animals**. For genetic labeling of chondrocyte lineage in control and *Gdf5* KO mice, *mTmG:Col2a1-Cre:Gdf5-CreER* control and mutant mice were used in all analyses. To generate *mTmG:Col2a1-Cre:Gdf5-CreER* mice, animals homozygous for *mTmg* (Jackson Laboratories) were crossed with *Col2a1-Cre* mice (Jackson Laboratories). Then, *mTmG:Col2a1-Cre* mice were crossed with mice heterozygous for *Gdf5-CreER*[74] creating control and *Gdf5* KO mice. Embryos were dissected in cold phosphate buffered saline (PBS), fixed overnight at 4 °C in 1% paraformaldehyde (PFA), washed in PBS and stored at 4 °C in 0.5 M EDTA (pH 8.0, Avantor Performance Materials) with 0.01% sodium azide (Sigma). In all timed pregnancies, the plug date was defined as E0.5. For harvesting of embryos, timed-pregnant female mice were sacrificed by $CO_2$ exposure. Embryos were sacrificed by decapitation with surgical scissors. Tails were visualized with fluorescent binoculars for genotyping when possible; alternatively, tail genomic DNA was used for genotyping by PCR (Supplementary Table 2). All animal experiments were pre-approved by and conducted according to the guidelines of the Institutional Animal Care and Use Committee (IACUC) of the Weizmann Institute. All animals used in this study had access to food and water ad libitum and were maintained under controlled humidity and temperature (45–65%, 22 ± 2 °C, respectively).

**Histology**. E18.5 or P6 tibias were fixed in 4% PFA overnight, dehydrated to 100% ethanol (dehydration sequence: 25%, 50%, 70%, and twice in 100% for 1 h each), and embedded in paraffin. 7-μm-thick paraffin sections were stained with H&E or Alcian Blue and collected onto slides. Slides were mounted with entellan and digitized with a Pannoramic scanner (3DHISTECH) and images were exported with QuPath software[92].

For immunofluorescence staining for pSMAD1/5/9, 7 μm-thick paraffin sections of embryo limbs ($n = 4$) were deparaffinized and rehydrated in water and stained as described in Felsenthal and Zelzer[93]. Slides were boiled in 0.3% Triton X-100/PBS for 10 min. Endogenous peroxidase was quenched by incubation in 2% $H_2O_2$/PBS for 30 min at room temperature. Next, staining was performed using primary anti-pSMAD1/5/9 antibodies (1:200, CST-13820, Cell Signaling Technology) followed by biotin-conjugated secondary antibodies (1:100, 711-065-152, Jackson ImmunoResearch) and HRP-conjugated streptavidin (1:200, NEL750001EA, PerkinElmer). Finally, detection was performed using Cy5-tyramide labeled fluorescent dyes, according to the instructions of the TSA Plus Fluorescent Systems Kit (PerkinElmer). Nuclei were counterstained with DAPI and sections were imaged with LSM 800 confocal microscope (Zeiss).

HCR smFISH[82] for *Gdf5* was performed on cryosections after 3 h fixation in 4% PFA/PBS (DEPC) followed by demineralization and dehydration overnight at 4 °C in 0.5 M EDTA/30% sucrose/PBS. The *Gdf5* probe was designed and ordered from (www.molecularinstruments.com), based on mRNA accession number NM_008109.3 [https://www.ncbi.nlm.nih.gov/nuccore/NM_008109.3], and amplified with B4 647 nm amplifier overnight. Finally, slides were counterstained using DAPI (1:1000, D9542, Millipore Sigma). Sections were imaged with LSM 800 confocal microscope (Zeiss) at a resolution of 312 nm (x,y). The brightness of the in situ signal was enhanced in Photoshop. For quantification of *Gdf5* transcripts per cell, amplification was performed for 1 h and *z*-stacks across the entire growth plate were acquired with a Zeiss LSM 800 confocal microscope at a resolution of 56 nm (x,y) and 380 nm (z). Autofluorescence imaging of cells was acquired with excitation at 488 nm laser. As a negative control, the same staining was performed

without the addition of the *Gdf5* probe. Quantification was performed on maximum intensity projection images after background subtraction (rolling ball radius 200 for cell boundary image and 50 for in situ signal) and mean filter (radius 2) for cell boundary images was performed in Fiji[94]. Cells were segmented with Cellpose[95], using the cyto algorithm with a diameter size between 75-500 pixels, depending on the area in the growth plate. The in situ signal was segmented in CellProfiler[96] with a custom pipeline (S6 data) to produce heatmaps of the number of transcripts per cell.

**Tissue clearing.** E16.5 tibias and ulnas from *mTmG:Col2a1-Cre:Gdf5-CreER* control and mutant mice were cleared using the PACT-deCAL technique[29,30]. Shortly, decalcified samples were washed in PBS, then embedded into a hydrogel of 4% (wt/vol) acrylamide in 1x PBS with 0.25% thermal initiator 2,2'-azobis[2-(2-imidazolin-2-yl)propane]dihydrochloride (Wako, cat. No. VA-044). The hydrogel was allowed to polymerize at 37 °C for 3 h. The samples were removed from the hydrogel, washed in PBS, and moved to 10% SDS with 0.01% sodium azide, shaking at 37 °C for 4 days, changing the SDS solution each day. Samples were washed four times with 1x PBST (PBS + 0.1% Triton X-100 + 0.01% sodium azide) at room temperature (RT) over the course of a day. To label nuclei, samples were submerged in 8 µg/ml DAPI in 1x PBST gently shaking overnight at RT. Samples were washed again with four changes of 1x PBST, and the refractive index (RI) of the sample was brought to 1.45 by submersion in a refractive index matching solution (RIMS) consisting of Histodenz (Sigma) and phosphate buffer, shaking gently at RT for 2-3 days. Samples were embedded in 1% low gelling agarose (Sigma) in PBS, in a glass capillary (Brand, Germany). Embedded samples were submerged in RIMS and protected from light at RT until imaging.

**Soft tissue micro-CT.** P12 control and *Gdf5* KO tibias were dissected, fixated overnight in 4% PFA in PBS, and dehydrated to 100% ethanol (dehydration sequence: 25%, 50%, 70%, and twice in 100% for 1 h each). According to ref. [97], samples were soaked in 2% iodine in 100% ethanol solution (Sigma) for 48 h at 4 °C. Tissue was washed in 100% ethanol twice for 30 minutes each prior to scanning. Samples were then mounted in 100% ethanol and scanned by MicroXCT-400 (Xradia) at 30 kV and 4.5 W with the Macro-70 lens.

**Light-sheet microscopy.** Cleared samples were imaged using a light-sheet Z1 microscope (Zeiss Ltd) equipped with 2 sCMOS cameras PCO- Edge, 10X illumination objectives (LSFM clearing 10X/0.2) and Clr Plan-Neofluar 20X/1.0 Corr nd=1.45 detection objective, which was dedicated for cleared samples in water-based solution of final RI of 1.45. A low-resolution image of the entire tibia was taken with the 20x Clarity lens at a zoom of 0.36. To acquire higher resolution images of the proximal and distal growth plates, multiview imaging was done with the same lens at a zoom of 2.5 resulting in x,y,z voxel sizes of 0.091, 0.091, 0.387 µm. The DAPI and Col2Cre-mGFP channels were acquired with Ex' 405 nm Em'BP 420-470 and Ex' 488 nm Em' BP 505-545 lasers at 2.4% and 3% laser powers, respectively. Light-sheet fusion of images was done in Zen software (Zeiss). Stitching of low-resolution images was done in ImarisStitcher.

**Growth plate segmentation.** Prior to segmentation, images were down-sampled in the X,Y direction using the Downsample plugin in Fiji[94] resulting in x,y,z voxel size of 0.194, 0.194, 0.387, and saved as TIFF files, which were then converted to VFF file format in MATLAB (S1 data). Non-cartilaginous regions were masked using Microview 2.1.2 (GE Healthcare). Shortly, spline contours were drawn around the growth plate throughout each z-stack, and the pixels outside the resulting 3D region of interest were reassigned an intensity value of 0 in Python (S2 data).

**Nucleus segmentation.** To automatically segment fluorescently labeled nuclei in the 3D images, we performed a two-step procedure. In the first step, seed points that were roughly located in the center of the nuclei were detected using a Laplacian-of-Gaussian-based (LoG) approach as described in[98]. In brief, the 3D input images were filtered with a LoG filter using a standard deviation that was empirically tuned to the radius of the objects of interest. We used standard deviations of $\sigma = 12$ for RZ, PZ and PHZ nuclei and of $\sigma = 35$ for HZ nuclei. Subsequently, local intensity maxima were extracted from the LoG-filtered image and reported as potential nuclei centers. For each potential seed point, we compute the mean intensity in a 4 x 4 x 4 voxel-wide cube surrounding the centroid. In order to minimize the number of false-positive detections, only seed points with a mean intensity larger than the global mean intensity of the entire LoG-filtered image were kept for further processing. In a final step, we used a seeded watershed algorithm to perform the segmentation of the nuclei in a Gaussian-smoothed ($\sigma = \sqrt{2}$) version of the intensity-inverted raw input image. The detected seed points were used to initialize the seeded watershed algorithm and we artificially added a background seed located at the border of each image snippet to separate the centered nucleus from the surrounding background.

The segmentation was performed separately for each nucleus and in parallel, i.e., small 3D image patches surrounding the seed points were processed concurrently using multiple cores of the CPU. Segmentation results of the

individually processed patches were then combined to form a full-resolution segmentation image containing the final result with a unique integer label for each of the nuclei that was used for further quantification and morphological analyses. All image analysis pipelines were implemented using the open-source software tool XPIWIT[61] and executed on a Windows Server 2012 R2 64-bit workstation with 2 Intel(R) Xeon(R)CPU E5-2690 v3 processors, 256 GB RAM, 24 cores and 48 logical processors (S3 data).

**Cell segmentation.** To segment fluorescently labeled cell membranes, we performed a manual contrast adjustment of the input images and extracted cell segments using a morphological watershed algorithm[99]. Owing to intensity variation in different regions of the image, we processed individual tiles separately and manually tuned the watershed starting level to obtain a good segmentation, i.e., all local minima below the specified level were used to initialize the catchment basins for the watershed regions. All image analysis pipelines were implemented using the open-source software tool XPIWIT[61] and executed on a Windows Server 2012 R2 64-bit workstation with 2 Intel(R) Xeon(R)CPU E5-2690 v3 processors, 256 GB RAM, 24 cores and 48 logical processors (S4 data).

**Nucleus and cell feature extraction.** Following image segmentation, images were resaved with multiple thresholds in order to avoid merging of neighboring objects during image binarization. Then, cell and nucleus volumes were measured using 3D manager[100] for each growth plate zone of each of the five growth plates, in order to set a minimal and maximal volume range to further clean the data (Supplementary Table 3). Finally, cells and nuclei overlapping the image borders were removed. To extract morphological features, each nucleus and cell were first converted from a binary volume into a triangulated mesh by applying Gaussian smoothing filter (sd = 0.5 pixels) and extracting the iso-surface at the iso-value 0.1. From the triangulated mesh of each object (nucleus or cell), the following features were extracted: surface area, as the sum of the areas of all mesh faces; volume, using the divergence theorem, also known as Gauss's theorem or Ostrogradsky's theorem[101]; sphericity, calculated as:

$$\Psi = \frac{\pi^{\frac{1}{3}}(6V_{\text{object}})^{\frac{2}{3}}}{A_{\text{object}}} \tag{1}$$

where $V_{\text{object}}$ is the volume of the object and $A_{\text{object}}$ is the surface area[102]; object orientation of each principal axis, defined as the direction of the first (PC 1), second (PC 2), or third (PC 3) principal components of the masked region of the object in real distance units. PC1 is the largest, PC2 the second largest, and PC3 is the smallest principal components. To calculate the cell bounding box height, the growth plate samples were registered such that the proximal-distal axis aligned to the global *z*-axis. Next, each cell was fit with an ellipsoid and a bounding box was fitted to the ellipsoid. The height of the bounding box along the proximal-distal axis is reported. Feature extraction was done on an Intel® Xeon® CPU E5-1620 v4 @ 3.50 GHz, 32.0 GB RAM, Windows 10 64 bit, with MATLAB version 2017b (The MathWorks, Inc., Natick, Massachusetts, USA).

**Growth plate registration.** All growth plates (PT, DT, and DU) were registered based on the outer surface of the RZ. For this, the mean of nuclear centroids were fixed to the origin of each growth plate sample. To remove bias due to heterogeneous density of centroids, only centroids located on the surface boundary points were used. Then, a bounding volume of each growth plate was extracted using the MATLAB function alphaShape polygons. MATLAB functions AlphaTriangulation and freeBoundary were used to extract the vertices from the outer surface of the growth plates and eigenvectors were calculated from the covariance of the vertices. This eigenvector matrix was used for the growth plate registration. We checked manually that distal, proximal, medial, lateral, dorsal and ventral parts of the RZ matched between growth plates. Manual multiplication of 180-degree rotation matrix was used to correct the eigenvectors if needed. The coordinates of the registered growth plates were calculated by multiplying the eigenvectors with the centroids.

**3D morphology maps of growth plates.** To visualize the spatial distribution of cellular or nuclear morphology within each growth plate, data were first stitched back together in MATLAB using the stage coordinates and scaling of each image. To place each voxel in a global coordinate system, we added the coordinates of each stack to every voxel in the stack, thereby reconstructing the entire bone. Growth plates were then registered based on manual alignment of the condyles in the RZ end of the growth plate. To highlight large patterns while averaging out small differences between individual nuclei, we represented nuclear and cellular features at a coarse-grained level. This representation was computed by first defining a regular spatial grid over the data. Each element of the grid was defined as a 75 × 75 x 75 µm cube containing up to 532 nuclei, and 255 cells and at least 178 nuclei and 119 cells. Within each cube, we averaged the nucleus and cell volume, surface area and sphericity and used the Delaunay tessellation field estimator at a resolution of 2 µm³ to compute the nuclear density for each cube[103]. Since most growth plate cells at E16.5 have only one nucleus, nuclear density was used to infer cell density as well. In addition, for each cube, we computed the occupation,

defined as the sum of nuclear or cellular volumes divided by the volume of the cube, N/C ratio and all other features as shown in Supplementary Table 1. Then, using MATLAB's jet color map, we represented the characteristics of each cube on the grid by drawing spheres whose radii and colors are proportional to the computed values and whose centers correspond to the average position of the nuclei or cells within the bin.

**Orientation maps**. The orientation of the three axes of a cell or a nucleus is described by the first, second, and third principal components of a PCA performed on the point cloud corresponding to the segmented shape of the cell or nucleus. When computing the principal component, the sign of the vector was attributed arbitrarily. Before averaging the orientations in a cube, we first constrained the orientations to be in the same hemisphere to avoid artificial bias. This was performed by considering all the orientations and their opposites, leading to $2N$ vectors, where $N$ is the number of objects in the cube. We then performed a PCA on the set of orientation vectors to determine the main direction. The $N$ orientation vectors corresponding to the hemisphere representing the main direction were extracted by considering only the ones having a positive dot product with the main direction. The main orientation was computed over the $N$ selected orientation vectors as a standard average over the direction cosines[104]. The spherical variance was computed as the dispersion around the mean direction[104]. For each cube, a line was drawn oriented and color-coded according to the deviation from the mean orientation of the whole growth plate, and whose thickness was proportional to the spherical variance. Finally, to summarize the distribution of orientations in a 3D map, we measured the difference of angle variation between mean orientation vectors obtained for each of the bins with mean orientation vectors obtained for the whole growth plate. A seven binned rainbow colormap (red to blue) was assigned to show small to large variations in angles.

**Spatial profiles of morphology features**. To visualize quantitatively the spatial profile of each of the computed features and identify gradients along the growth plate, the entire growth plate was divided into 50 equally spaced bins along the P–D axis. For each bin, the mean value of individual cell or nucleus features were computed from the cells/nuclei whose centroid fell in this bin. Finally, bins were plotted with the x-axis representing the P-D axis and y-axis representing the mean value of the feature among all samples for each bin.

**t-test of the means and standard deviations of morphological features**. To detect significantly different regions along the P-D axis of two groups of growth plates, a two-tailed t-test on the spatial profile of a given feature was performed. The regions were tested by two ways, first using the mean value of features and second using the standard deviation of features between groups; i.e., five control and four mutant samples. p-value ≤ 0.05 indicates statistical significance. The t-tests were performed as follows: We chose the window length of 5 and checked for the significant regions in the window of bins $[i$ to $i + 5]$, where $i$ index ranges from 1 to 45. All the windows the bins of which had $p \leq 0.05$ were merged and the final p-value was recalculated on these merged bins, showing the significant regions in the graph.

**Linear regression of vol$^{2/3}$/SA graphs**. Four linear regressions were fit to data in RZ, PZ, PHZ, and HZ using the MATLAB function POLYFIT. The boundary between zones in the x-range was adjusted manually, such that each growth plate used the same range of data. The slope and intercept value of the linear regression are mentioned in the figure or legend (Figs. 3 and 6) and the fitted line was plotted using POLYVAL function. If actual and fitted data is y and yfit, respectively, then coefficient of determination $R^2$ is

$$R^2 = 1 - \frac{\sum_{i=1}^{N}(y_i - yfit_i)^2}{\sum_{i=1}^{N}(y_i - \bar{y})^2} \qquad (2)$$

and standard error of slope coefficient is

$$SE = \frac{\sqrt{\frac{\sum_{i=1}^{N}(y_i - yfit_i)^2}{N-2}}}{\sqrt{\sum_{i=1}^{N}(x_i - \bar{x})^2}} \qquad (3)$$

p-value of the slope was calculated using Student's t-test cumulative distribution function

$$2*tcdf(abs(slope/SE), dof, 'upper') \qquad (4)$$

where the slope is the slope coefficient and dof is the degrees of freedom in the given data.

**Reporting summary**. Further information on research design is available in the Nature Research Reporting Summary linked to this article.

## Data availability

The datasets generated and analyzed during the current study are available from the corresponding author on reasonable request. Additionally, source data are provided

within this manuscript and 3D MAPs sample data to test the codes are available on Figshare with the identifiers [https://doi.org/10.6084/m9.figshare.14903052.v1] and [https://doi.org/10.6084/m9.figshare.14932503.v1]. Source data are provided with this paper.

## Code availability

The reviewed version of 3D MAPs and preprocessing codes are available on Zenodo with the identifiers [https://zenodo.org/badge/latestdoi/352405998] and [https://zenodo.org/badge/latestdoi/352588343].

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

## Acknowledgements

We thank Nitzan Konstantin for editorial assistance and members of the Zelzer lab for their advice and encouragement throughout this project. We thank the de Picciotto Cancer Cell Observatory in memory of Wolfgang and Ruth Lesser, Weizmann Institute of Science, for providing LSFM infrastructure, Rada Massarwa and Jacob Hanna from the Department of Molecular Genetics, Weizmann Institute, for providing *mTmg* mice, Ofra Golani from the MICC Cell Observatory Weizmann Institute, for support and discussions regarding segmentation analysis, Ron Rotkopf from the Bioinformatics Unit for consulting us on statistics, Tali Wiesel from the Graphic Design Department at the Weizmann Institute of Science for her help with graphics, and Chinami Michaels for her help creating illustrations of our findings. This study was supported by grants from the David and Fela Shapell Family Center for Genetic Disorders and by The Estate of Mr. and Mrs. van Adelsbergen (to E.Z), the WIN program between Princeton University and the Weizmann Institute (to P.V.), and from the German Research Foundation (DFG-MI1315/4-1) to J.St.

## Author contributions

S.R. designed and carried out experiments and analyses and wrote the manuscript; T.S., P.V., and A.A. designed and carried out analyses; J. Stegmaier designed analyses; J. Svorai assisted with data preprocessing; S.K. carried out immunohistochemistry and in situ experiments; N.F. designed HCR quantification pipeline; Y.A. assisted with benchmark imaging; E.Z. designed and supervised experiments and analyses and wrote the manuscript. All authors reviewed the manuscript.

## Competing interests

The authors declare no competing interests.
