## [Peer Review File · Nature Communications]

REVIEWER COMMENTS

Reviewer #1 (Remarks to the Author):

Abstract

1. The abstract does not explain what 3D MAPs is.
2. What does orthologous mean here? When used to refer to genes, orthologous implies the corresponding genes in two different species.

Results

3. It would be helpful to show a (2-dimensional) image of a resting, proliferative, pre-hypertrophic and hypertrophic chondrocyte using the methodology described. That would allow the reader to understand better the resolution. It would be preferable to show representative (not unusually good) images.
4. Measurements included cell volume and surface area. I would suggest analyzing the diameter of the hypertrophic cells in the proximal-distal dimension because there is reason to think that this diameter of the terminal hypertrophic chondrocyte is a critical determinant of the rate of longitudinal bone growth (Kember NF, Walker KV. Control of bone growth in rats. *Nature*. 1971; 229(5284):428-9. PMID: 4927000; Sissons HA. Experimental study on the effect of local irritation on bone growth. In: Mitchell JSH, B.F.; Smith C.L., editor. *Progress in Radiobiology, Proceedings of the Fourth International Conference on Radiobiology*. Edinburgh and London: Oliver & Boyd; 1955. p. 436-48).
5. The authors discuss their findings in terms of the concept that "There is a well-established positive correlation between hypertrophic cell volume and growth potential of different growth plates." This vague concept has crept into the recent literature, replacing a more specific and precise concept that the rate of longitudinal bone growth is approximately equal to the product of the number of chondrocytes produced per column per unit time, multiplied by the length that each of those chondrocytes eventually contributes to longitudinal growth i.e. the proliferation rate per column multiplied by the height of the terminal hypertrophic chondrocyte (same references as above). I would urge the authors to reassess their findings in light of this powerful concept.
6. The results subsection entitled, "Allometric and isometric growth behaviors define chondrocyte shape throughout differentiation," contains little novel information. It has been known for decades that the resting zone chondrocytes are something like spheres, the proliferative zone chondrocytes are flattened like pancakes, and the hypertrophic zone chondrocytes lose that flattening. Thus, the diameter parallel to the proximal-distal axis of the bone decreases than increases. This subsection seems to restate this well-known concept using the terms isometric and allometric, which, in my view, is an unnecessarily convoluted way to view the straightforward geometric changes. Of course, a sphere will have a greater volume vs surface area than a short cylinder (pancake), but the changes are far easier to conceptualize in terms of simple changes in shape than in terms of $Vol^{2/3} / SA$. Thus the subsection has two problems. First it contains little new information, and second it obfuscates.
7. The finding that growth plate chondrocytes are longer in the dorsal-ventral axis than in the medial-lateral axis is interesting, although, as the authors acknowledge, the biological function and implications are unclear. However, such novel findings require strong evidence. I think the possibility of an artifact arising from the technique needs to be excluded. Could the asymmetry in the medial-lateral diameter vs the dorsal-ventral diameter be a false result arising from the asymmetry in the measurement approach? Was the light sheet microscopy always done in the plane defined by the proximal-distal axis and the medial-lateral axis? Or was it always done in the plane defined by the proximal-distal axis and the dorsal-ventral axis? If so, I suggest repeating the approach using light sheet microscopy in the other plane and also in the plane defined by the medial-lateral axis and the dorsal-ventral axis i.e. a cross-section of the bone. Alternatively, or additionally, the asymmetry in the diameters might be verified by confocal microscopy or simply by measuring cell diameters in histological sections made in various planes.
8. The 3D MAPs approach provided a precise description of how Gdf5 knockout affects cell size, shape, and orientation. However, it's not clear to me that such precise geometric descriptions provide much insight into the underlying molecular-biology and cell-biology mechanisms by which lack of Gdf5 protein affects chondrocyte function, aside from the simple observation that the hypertrophic chondrocyte does not grow as large.
9. In general, the 3D MAPs approach looks like a valuable tool to describe cell size, shape, and

orientation throughout the growth plate. However, I am not convinced that the resulting detailed geometric descriptions provide much mechanistic insight. The diameter of the terminal hypertrophic chondrocyte measured in the proximal-distal dimension is a key variable in determining the rate of longitudinal growth and thus is important to measure. But I do not see great importance of the detailed measurements of proliferative zone or resting zone chondrocytes. Detailed geometric measurements do not, for example, tell us how cell size is governed by the cytoskeleton, by fluid shifts, or by synthesis of specific cellular components. Nor do such measurements tell us the molecular pathways by which a genetic abnormality affects the cell.

Reviewer #2 (Remarks to the Author):

The paper present original and promising results on chondrocytes morphology in different regions of the epiphyseal growth plate of long bones using tissue clearing and light sheet microscopy. An original image processing tool called 3D MAPs is proposed to explore a large dataset of cells with a multiscale approach from the cell level to the tissue level. This tool was used to investigate the cell growth, and explore the GDF5 regulation of chondrocytes, by comparing Gdf5 KO growth plate to control, with rather convincing results, although still preliminary.

However, despite its originality and the important amount of work that was put in the paper, there are several concerns requiring a major revision of the paper:

1/ the paper was difficult to read because it is not organized in the standard Introduction -M&M- results-discussion scheme : results in page 5 come just after introduction, then data acquisition and segmentation in page 7, then a piece of conclusion in page 8 (3D MAP scan identify morphogenetic behaviors in the growth plates), then some mix between M&M and results from pages 8 to 23, then discussion from page 23 to 27, then methods part from 27 to 35. A lot of methodological information is provided either in the last part or in supplementary materials. As an example Tibiae and ulnae are used, but number of specimens is provided at the end ; we find it in the legends of supplementary figures (page 46 to 48): it seems from supplementary figure 1 legend that the sample size is 20 growth plates, from the figure 2 supplementary legend that the numbers of samples is 5 for proximal and distal tibia, and 3 for ulna, and from the supplementary figure 3 legend that the comparison of control and mutants is performed on tibiae of 5 control and 4 mutants. Also the legends are very long, often exceeding the recommended maximum of 350 words, and contain M&M information that could be in the core of the text.

2/ the paper is very long because it present three messages : the description and validation of the 3D MAPs tool, the use of this tool to explore the chondrocyte shape throughout differentiation, and the regulatory role of GDF5 in the growth plate process. Perhaps this could be split into two papers (one more technical on 3D MAPs tool and the other on the chondrocyte morphometry).

Alternatively the data acquisition and segmentation could be described in detail in a technical annex, with full description and validation of the process, therefore focusing the paper on the application issues.

3/ the 3D MAPs approach is attractive, but the validation is insufficient : in supplementary figure 1 D comparing manual and automatic processing, we see that differences can reach 30% for cell numbers of a given volume or surface area, to 50% for nuclei numbers of a given volume or surface area: The fact that there is no significant difference between manual and automatic methods is not sufficient to « assess validity » (see for example Bland & Altman, Statistical methods for assessing agreement between two methods of clinical measurement Lancet 1986).

What is the limit of agreement between the two methods, and what is the impact of such differences ?

Also, from automatic processing, 17 parameters are explored related to the cells shape and orientation, what is the uncertainty of such measurements , and what is the impact of such uncertainties on the conclusions ? for example in comparing Gdf5 KO tibiae to controls (page 19) a 8% difference was found for sphericity : is this difference in the margin of uncertainty ?

4/ Finally, in scientific papers, the authors generally acknowledge and discuss the limitations of their study, and this part is missing : for example the sample size is quite limited, is it sufficient to draw conclusions as regards the difference Gdf5 KO tibiae to controls, given the variability among individuals ?

REVIEWER COMMENTS

Reviewer #1 (Remarks to the Author):

Abstract

1. The abstract does not explain what 3D MAPs is.

As requested, we added a description of the acronym 3D MAPs in the abstract.

2. What does orthologous mean here? When used to refer to genes, orthologous implies the corresponding genes in two different species.

Orthologous was changed to zeugopodal, to describe the relationship between the tibia and ulna.

Results

3. It would be helpful to show a (2-dimensional) image of a resting, proliferative, pre-hypertrophic and hypertrophic chondrocyte using the methodology described. That would allow the reader to understand better the resolution. It would be preferable to show representative (not unusually good) images.

As requested by the reviewer, we added 2D images in Supplemental figure 1A. These images represent the high quality data that we normally get with light sheet microscopy on cleared E16.5 growth plates.

4. Measurements included cell volume and surface area. I would suggest analyzing the diameter of the hypertrophic cells in the proximal-distal dimension because there is reason to think that this diameter of the terminal hypertrophic chondrocyte is a critical determinant of the rate of longitudinal bone growth (Kember NF, Walker KV. Control of bone growth in rats. *Nature*. 1971; 229(5284):428-9. PMID: 4927000; Sissons HA. Experimental study on the effect of local irritation on bone growth. In: Mitchell JSH, B.F.; Smith C.L., editor. *Progress in Radiobiology, Proceedings of the Fourth International Conference on Radiobiology*. Edinburgh and London: Oliver & Boyd; 1955. p. 436-48).

Following this comment, we added this measurement to the manuscript. Comparing the height of the hypertrophic chondrocyte (Figure L1 below) to the cell volume (Figure 2C) allows not only to determine the correlation between the two measurements, but also to evaluate their predictive value with regard to longitudinal bone growth. For both measurements, we observed similar trends of increasing in the PHZ and peaking in the HZ. However, we found that cell diameter was correlated only with some of the differences in bone growth, whereas the volume of hypertrophic cells was correlated with all the observed differences (Figure 2B and Supplementary figure 3A and B). This suggests that cell volume is a better predictor of bone elongation than cell diameter. This finding is in agreement with previous studies (Breur, Vanenkevort et al. 1991, Wilsman, Farnum et al. 1996, Breur, Md. et al. 1997, Farnum, Lee et al. 2002, Wilsman, Bernardini et al. 2008, Cooper, Oh et al. 2013, Cooper 2019).

Figure L1. Cell bounding box height along the P-D axis. Spatial profiles of the mean cell bounding box height from the resting zone through to the end of the hypertrophic zone. (A) Comparison between WT distal and proximal tibias shows that the mean cell height in the proximal tibia is significantly larger in the transition between the resting and proliferative zone ($p < 0.01$). (B) Comparison between WT distal ulna and proximal tibia shows that the mean cell height in the proximal tibia is significantly larger in the resting zone and that of the distal ulna is significantly larger in the hypertrophic zone ($p < 0.01$). (C) Comparison between WT distal ulna and distal tibia shows that the mean cell height in the distal ulna is significantly larger in the proliferative and hypertrophic zones ($p < 0.01$). (D) Comparison between control and *Gdf5* KO (mut) proximal tibias shows that the mean cell height in the mutant is significantly larger in the resting ($p < 0.01$) and proliferative zones ($p < 0.05$). (E) Comparison between control and *Gdf5* KO (mut) distal tibias shows that the mean cell height in the mutant is significantly larger in the resting ($p < 0.05$), proliferative ($p < 0.01$), and hypertrophic zones ($p < 0.05$). (F) Illustration depicting how the height of the cell bounding box along the P-D axis is influenced by the orientation of the cell within the growth plate. p-values were calculated by Student's *t*-test between mean value of proximal tibia control ($n = 5$), distal tibia control ($n = 5$), distal ulna control ($n = 3$), and proximal and distal tibia mutant samples ($n = 4$).

5. The authors discuss their findings in terms of the concept that “There is a well-established positive correlation between hypertrophic cell volume and growth potential of different growth plates.” This vague concept has crept into the recent literature, replacing a more specific and precise concept that the rate of longitudinal bone growth is approximately equal to the product of the number of chondrocytes produced per column per unit time, multiplied by the length that each of those chondrocytes eventually contributes to longitudinal growth i.e. the proliferation rate per column multiplied by the height of the terminal hypertrophic chondrocyte (same references as above). I would urge the authors to reassess their findings in light of this powerful concept.

We agree with the reviewer that “the rate of longitudinal bone growth is approximately equal to the product of the number of chondrocytes produced per column per unit time, multiplied by the length that each of those chondrocytes eventually contributes to longitudinal growth”. However, to apply this concept experimentally, it is necessary to recognize a column in the growth plate. In contrast to the mature growth plate, where a clone of cells in the proliferative zone forms a clear column that gives rise to the hypertrophic column, this linear relationship does not exist in the embryo, where the column structure is immature. Moreover, a recent report in *Nature* (Newton, Li et al. 2019) shows major differences between column formation in embryonic vs postnatal growth plates. Embryonic columns are multiclonal, breaking the linearity that characterizes postnatal columns. Indeed, this formula was used only in postnatal bones, and was never proven to predict potential bone growth in the embryo (HA. 1955, Kember and Walker 1971, Lui, Jee et al. 2018). Therefore, we analyzed the widely accepted criterion of hypertrophic cell volume in growth plates that are known to differ in potential growth (Breur, Vanenkevort et al. 1991, Wilsman, Farnum et al. 1996, Farnum, Lee et al. 2002, Wilsman, Bernardini et al. 2008, Cooper, Oh et al. 2013). We showed that the HZ cell volume was sensitive enough to predict these differences in growth, as well as predict the growth retardation in *Gdf5* KO bones (Figure 2C and 6A). Finally, since we believe that 3D MAPs can be used not only in embryonic bones, following the reviewer’s comments we added to the discussion a paragraph suggesting the use of the formula mentioned by the reviewer in postnatal bones.

6. The results subsection entitled, “Allometric and isometric growth behaviors define chondrocyte shape throughout differentiation,” contains little novel information. It has been known for decades that the resting zone chondrocytes are something like spheres, the proliferative zone chondrocytes are flattened like pancakes, and the hypertrophic zone chondrocytes lose that flattening. Thus, the diameter parallel to the proximal-distal axis of the bone decreases than increases. This subsection seems to restate this well-known concept using the terms isometric and allometric, which, in my view, is an unnecessarily convoluted way to view the straightforward geometric changes. Of course, a sphere will have a greater volume vs surface area than a short cylinder (pancake), but the changes are far easier to conceptualize in terms of simple changes in shape than in terms of $Vol^{2/3} / SA$. Thus the subsection has two problems. First it contains little new information, and second it obfuscates.

First, the reviewer points to the relative complexity arising from a quantitative description of the shapes of cells, suggesting that concepts such as isometric and allometric growth are less intuitive than qualitative terms such as “spheres” or “flattened pancakes”. Second, the reviewer

suggests that the information presented in this subsection, which provides detailed quantitative characterization of growth patterns in the growth plate, is unclear. We might have failed to explain the value of transforming qualitative characteristics of cell morphogenesis into quantitative 3D measurements and present the wealth of novel findings we obtained using 3D MAPs. To clarify these issues, we made textual changes in the manuscript and, in the following, we will try to explain the novelty of our findings.

Bone elongation is the output of sequential morphogenesis of chondrocytes and, thus, any newly described cellular behaviors can likely play a role in bone morphogenesis. The first finding that chondrocytes change their growth strategy between allometric and isometric growth when they differentiate and enter a new zone shows that during chondrocyte differentiation, cell morphology is sculpted through axis-specific growth only in the first three zones (Figure 3A). Once cells enter the hypertrophic zone, their shape is already determined, and they only change their size through isometric growth. The advantage of quantitative analysis of cell growth is shown by the ability to distinguish between two strategies of allometric growth, namely surface-area dependent and volume-dependent. Whereas surface area-dependent growth occurring during the flattening of proliferative chondrocytes implies regulation by the cytoskeleton, volume-dependent growth occurring during the swelling of resting and prehypertrophic chondrocytes implicates mechanisms involved in rapid water uptake. Thus, creating a roadmap for when and where chondrocytes use directional growth to change their morphology, we discovered that the repertoire of morphological changes is much broader than previously thought, which is likely important for bone growth.

A testimony to the significance of the morphogenetic switch between allometric and isometric growth is the observation that resting zone and hypertrophic chondrocytes have the same aspect ratios, i.e. the same shape (Figure 3C). This means that theoretically, the cells could have grown isometrically throughout the growth plate, but they choose not to. One possible reason for that is that due to the different density and packing within the different zones, isometric growth throughout the growth plate would cause a much wider tissue. Therefore, the changes in cell growth mechanisms serve not only to determine cell morphologies in each zone, but also likely to sculpt the shape of the growth plate itself.

Finally, the ability to differentiate between strategies is important not only to understand the growth process, but also to identify when it is abnormal. This is exemplified in our study of *Gdf5* KO growth plates. Although it was long known that *Gdf5* KO bones were shorter, there was little understanding of what goes wrong with these bones. Our work shows that while hypertrophic chondrocytes grow normally, they are significantly smaller than controls because of aberrant growth in the previous zones. This indicates that the morphologic changes occurring during early stages of chondrocyte differentiation have a cumulative effect and, therefore, to fully understand the output of bone elongation, it is necessary to document changes that occur in all growth plate zones. Additionally, this finding suggests that GDF5 is involved in the regulation of surface area- and volume-dependent allometric growth (Figure 6A and 6C).

These novel discoveries promote the field in several ways. First, they provide a quantitative description of cell morphogenesis, which can be used as a ground truth for future studies. Second, they expose the need for yet unknown molecular mechanisms to regulate the various

morphogenetic processes. Third, they show that column formation and chondrocyte hypertrophy are not the only morphologic changes that affect bone elongation. Finally, they allow a better understanding of different mutations and thereby of growth plate biology and disease.

7. The finding that growth plate chondrocytes are longer in the dorsal-ventral axis than in the medial-lateral axis is interesting, although, as the authors acknowledge, the biological function and implications are unclear. However, such novel findings require strong evidence. I think the possibility of an artifact arising from the technique needs to be excluded. Could the asymmetry in the medial-lateral diameter vs the dorsal-ventral diameter be a false result arising from the asymmetry in the measurement approach? Was the light sheet microscopy always done in the plane defined by the proximal-distal axis and the medial-lateral axis? Or was it always done in the plane defined by the proximal-distal axis and the dorsal-ventral axis? If so, I suggest repeating the approach using light sheet microscopy in the other plane and also in the plane defined by the medial-lateral axis and the dorsal-ventral axis i.e. a cross-section of the bone. Alternatively, or additionally, the asymmetry in the diameters might be verified by confocal microscopy or simply by measuring cell diameters in histological sections made in various planes.

As suggested by the reviewer, to verify that the imaging angle does not introduce an artifact along the z axis, we analyzed an entire ulna imaged at two orthogonal rotations along the P-D axis, such that the z axis was along either the dorsal-ventral or medial-lateral axis of the bone. As seen in Figure L2A below, the orientation of PC1 was not changed between the two imaging angles, ruling out the possibility for an artifact. It is important to note that the two images are not completely identical, as one might expect. This is because in light sheet microscopy, the optimal imaging angle is the one with the sharpest signal and that creates the shortest distance for the sheet to penetrate through the sample. The differences we see in our sample are because one angle is free from surrounding tissues, whereas the other angle is not, creating a difference in light passage. As a result, one angle produces better quality images for segmentation.

As suggested by Reviewer 2, to quantitatively show that the two images are identical, we performed a segmentation error analysis. Results show that the expected error in PC1 cell orientation is 6° - 16° , depending on the growth plate zone (Figure L5). Therefore, if we had an imaging artifact, we would expect differences of 74° - 90° between the two samples. However, we found that PC1 orientations were mostly within the range of 10° - 40° , showing that there is no artifact.

A PC1 cell orientation imaged along orthogonal axes

B Distal Ulna registration

Figure L2. Comparison of PC1 cell orientation between two orthogonal imaging angles. (A) 3D morphology map of PC1 cell orientation of the same distal ulna sample imaged by light sheet microscopy at two orthogonal rotations (0°, dorsal-ventral axis; 90°, medial-lateral axis) shows high similarity between the two imaging rotations. (B) Two different views of the registered data (red, dorsal-ventral; blue, medial-lateral) show that they do not perfectly overlap (View 1). Black arrows indicate regions of low overlap that have the highest difference in PC1 cell orientation in panel A.

8. The 3D MAPs approach provided a precise description of how *Gdf5* knockout affects cell size, shape, and orientation. However, it's not clear to me that such precise geometric descriptions provide much insight into the underlying molecular-biology and cell-biology mechanisms by which lack of *Gdf5* protein affects chondrocyte function, aside from the simple observation that the hypertrophic chondrocyte does not grow as large.

We agree with the reviewer that studying geometric measurements does not by itself elucidate a molecular mechanism. Nevertheless, 3D MAPs serves two roles. The first is to recover the morphological sequence chondrocytes undergo during differentiation, which is in fact the cell-biology mechanism driving the morphogenesis of the bone. The second is to identify abnormal cellular behaviors, which creates the need for molecular mechanisms to regulate the process.

In regard to our analysis of *Gdf5* mutants, prior to our study, based on classical 2D histological techniques, *Gdf5* KO growth plates were thought to be normal (Mikic, et al. 2004). Using 3D MAPs, we discovered a collection of cellular abnormalities that start in the RZ and relate to cell growth, morphology, and organization. These results reveal both the sensitivity of 3D MAPs as well as the complexity of the phenotype. The numerous abnormalities we found throughout the different growth plate zones in the *Gdf5* KO bones highlight the significance of analyzing the entire growth plate, and not only hypertrophic chondrocytes. Most notably, we found that although mutant hypertrophic cells were 30% smaller in volume (Figure 6A), their isometric growth was unaffected (Figure 6C) and the volume deficiency arose from abnormal growth in the PZ and PHZ.

To accommodate the request for mechanistic insights, we studied *Gdf5* expression at high resolution using single-molecule fluorescent in situ hybridization. Results in the revised manuscript and below show that *Gdf5* is expressed throughout the growth plate (Figure L3A-H below and Figure 5 A-H). Previously, *Gdf5* was known to be expressed only in the joint and outer collar region (Storm, T.V. et al. 1994, Storm and Kingsley 1996, Settle, Rountree et al. 2003, Chen, Capellini et al. 2016, Shwartz, Viukov et al. 2016, Capellini, Chen et al. 2017). To understand the significance of this newly discovered *Gdf5* expression in the growth plate, we studied the activity of its downstream effectors Smad 1/5/9 (Figure L3I and Figure 5I). Interestingly, we identified reduced phosphorylation of Smads in the *Gdf5* KO growth plate, which might explain some of the phenotypes we observed. Obviously, this is only the beginning of understanding the role of GDF5 in growth plate biology and many questions still need to be answered, such as whether the effect is autonomous or not, and how Smad activity regulates different aspects of cell morphology.

Figure L3. *Gdf5* expression and its effect on Smad activity. smFISH for *Gdf5* in the proximal tibia (A) and knee joint (Fe, femur; Ti, tibia; Fi, fibula) (B). Scale bars: 200 μ m. *Gdf5* is expressed in the collar region (C), resting zone (D), proliferative zone (E), prehypertrophic zone (F), and hypertrophic zone (G) (n = 4). Scale bars: 10 μ m. (H) A heat map shows the number of *Gdf5* transcripts per cell throughout the proximal tibia growth plate (n = 3). The highest signal per cell is in the collar region, followed by the proliferative zone adjacent to the collar, and some areas in the resting zone. Notably, most cells have between 0-1 transcripts. (I) Immunofluorescence for pSMAD 1/5/9 shows a reduction in the proximal tibia of *Gdf5* KO mice compared to controls (n = 5). Scale bar: 200 μ m. Blue and yellow square insets show zoomed-in regions in the resting and proliferative zones, respectively. Scale bar: 25 μ m.

9. In general, the 3D MAPs approach looks like a valuable tool to describe cell size, shape, and orientation throughout the growth plate. However, I am not convinced that the resulting detailed geometric descriptions provide much mechanistic insight. The diameter of the terminal hypertrophic chondrocyte measured in the proximal-distal dimension is a key variable in determining the rate of longitudinal growth and thus is important to measure. But I do not see great importance of the detailed measurements of proliferative zone or resting zone chondrocytes. Detailed geometric measurements do not, for example, tell us how cell size is governed by the cytoskeleton, by fluid shifts, or by synthesis of specific cellular components. Nor do such measurements tell us the molecular pathways by which a genetic abnormality affects the cell.

We appreciate the reviewer's recognition of the value of 3D MAPs as a tool to describe cell size, shape, and orientation throughout the growth plate. Regarding the reviewer's concerns, the main function of the growth plate is to drive growth and morphogenesis of the future bone. It performs this role through sequential change in morphology of cells in each zone. It has been clearly shown in a variety of organisms and tissues that the 3D morphology of cells is a major driving force for tissue morphogenesis (Lewis 1931, Warga and Kimmel 1990, Shih and Keller 1992, Shih and Keller 1992, Irvine and Wieschhaus 1994, Hogan 1999, Lawson and Schoenwolf 2001, Fu, Gu et al. 2005, Farhadifar, Röper et al. 2007, Lecuit and Lenne 2007, Metzger, Klein et al. 2008, Martin, Kaschube et al. 2009, Paluch and Heisenberg 2009, Bénazéraf, Francois et al. 2010, Kim and Davidson 2011, Blanc, Coste et al. 2012, Short, Combes et al. 2014, Armour, Barton et al. 2015, Bénazéraf, Francois et al. 2010, Bénazéraf, Beaupeux et al. 2017, Carter, Sánchez-Corrales et al. 2017, Lefevre, Short et al. 2017). Therefore, the only way to understand tissue morphogenesis is to first learn the morphogenetic behaviors of cells. This requires a reliable 3D tool that allows accurate quantification of these behaviors, which are also the mechanism by which bones grow.

The motivation for creating 3D MAPs was our belief that a wealth of morphological phenomena in the growth plate, which might play a role in its activity, had not yet been described. Indeed, 3D MAPs revealed several new morphological characteristics, including the diversity of chondrocyte growth patterns and how it produces a variety of specific morphologies (Figure 3), as well as the hierarchy and dynamics of cell axis orientations (Figure 4). These discoveries can now serve us and others in the community as a basis for identifying molecular mechanisms that regulate these phenomena.

The reviewer claims that "detailed geometric measurements do not ... tell us how cell size is governed by the cytoskeleton, by fluid shifts, or by synthesis of specific cellular components." We believe that detailed geometric measurements can provide valuable information about the underlying regulatory mechanisms. For example, we found that chondrocytes change their morphology by employing different strategies of allometric growth. While cell flattening, which is driven by surface area, implies regulation by the cytoskeleton, volume-driven cell swelling implicates mechanisms involved in rapid water uptake (Figure 3A). Another example is our finding that volume dependent allometric growth is 50% slower in the prehypertrophic zone of *Gdf5* KO growth plates. This suggests that TGF β signaling regulates components of the cytoskeleton or ion channels through GDF5, which would normally allow rapid volume

enlargement in this zone (Figure 6C). Without the geometric descriptions, the potential association between the TGF β family and these cellular machineries would remain hidden.

Regarding the reviewer's statement that the diameter of hypertrophic chondrocytes in the P-D dimension is important to measure, as mentioned, we show that cell volume is more reliable than the diameter of hypertrophic chondrocytes in the P-D dimension in predicting the extent of bone elongation. This shows the importance and value of 3D measurements (Figure L1 and Figure 2C). Second, regarding the reviewer's statement that they "do not see the importance of the detailed measurements of proliferative zone or resting zone chondrocytes", we believe that our findings suggest otherwise. While volume increase of hypertrophic chondrocytes plays a major role in bone elongation, our study provides several clear examples of how changes that occur before these cells reach the hypertrophic zone affect growth. As we mention before, for example, although hypertrophic chondrocytes in *Gdf5* KO growth plates grow normally, their final volume is 30% smaller because they carry a 50% growth deficit from the prehypertrophic zone (Figure 6A and 6C). Moreover, we show that 50% of the total growth of chondrocytes occurs in the prehypertrophic zone, as compared to 35% in the hypertrophic zone (Figure 2A). Interestingly, Li, Trivedi et al. (2015) used live imaging of chick embryonic growth plates to show that bone elongation was driven by cell volume enlargement in the proliferative zone. These are clear demonstrations of the essential contribution of morphogenetic processes occurring before the hypertrophic zone and, hence, of the importance of quantitative studies of the entire growth plate, and not just the hypertrophic zone.

In conclusion, this work provides the first high-resolution mapping of chondrocyte morphogenesis in an intact growth plate. The use of detailed geometric measurements has shown that the landscape of morphological changes is much broader than column formation and hypertrophy. These findings not only show that chondrocyte morphogenesis is more complex than previously thought, but also open many new questions on the mechanisms that regulate it. We hope that we were able to convince Reviewer 1 of the significance and novelty of our work.

Reviewer #2 (Remarks to the Author):

The paper present original and promising results on chondrocytes morphology in different regions of the epiphyseal growth plate of long bones using tissue clearing and light sheet microscopy. An original image processing tool called 3D MAPs is proposed to explore a large dataset of cells with a multiscale approach from the cell level to the tissue level. This tool was used to investigate the cell growth, and explore the GDF5 regulation of chondrocytes, by comparing *Gdf5* KO growth plate to control, with rather convincing results, although still preliminary.

However, despite its originality and the important amount of work that was put in the paper, there are several concerns requiring a major revision of the paper:

1/ the paper was difficult to read because it is not organized in the standard Introduction -M&M- results-discussion scheme : results in page 5 come just after introduction, then data acquisition and segmentation in page 7, then a piece of conclusion in page 8 (3D MAP scan identify morphogenetic behaviors in the growth plates), then some mix between M&M and results from pages 8 to 23, then discussion from page 23 to 27, then methods part from 27 to 35. A lot of methodological information is provided either in the last part or in supplementary materials. As

an example Tibiae and ulnae are used, but number of specimens is provided at the end ; we find it in the legends of supplementary figures (page 46 to 48): it seems from supplementary figure 1 legend that the sample size is 20 growth plates, from the figure 2 supplementary legend that the numbers of samples is 5 for proximal and distal tibia, and 3 for ulna, and from the supplementary figure 3 legend that the comparison of control and mutants is performed on tibiae of 5 control and 4 mutants. Also the legends are very long, often exceeding the recommended maximum of 350 words, and contain M&M information that could be in the core of the text.

As suggested by the reviewer, we have reorganized the paper by placing the technical explanations of the method in a supplement. Additionally, we have corrected the sample numbers in the text.

2/ the paper is very long because it present three messages : the description and validation of the 3D MAPs tool, the use of this tool to explore the chondrocyte shape throughout differentiation, and the regulatory role of GDF5 in the growth plate process. Perhaps this could be split into two papers (one more technical on 3D MAPs tool and the other on the chondrocyte morphometry). Alternatively the data acquisition and segmentation could be described in detail in a technical annex, with full description and validation of the process, therefore focusing the paper on the application issues.

To sharpen the focus of the paper, we put the data acquisition and segmentation in a technical annex, with full description and validation of the process.

3/ the 3D MAPs approach is attractive, but the validation is insufficient : in supplementary figure 1 D comparing manual and automatic processing, we see that differences can reach 30% for cell numbers of a given volume or surface area, to 50% for nuclei numbers of a given volume or surface area: The fact that there is no significant difference between manual and automatic methods is not sufficient to « assess validity » (see for example Bland & Altman, Statistical methods for assessing agreement between two methods of clinical measurement Lancet 1986). What is the limit of agreement between the two methods, and what is the impact of such differences ?

Also, from automatic processing, 17 parameters are explored related to the cells shape and orientation, what is the uncertainty of such measurements , and what is the impact of such uncertainties on the conclusions ? for example in comparing Gdf5 KO tibiae to controls (page 19) a 8% difference was found for sphericity : is this difference in the margin of uncertainty ?

To evaluate the limit of agreement and impact of differences between segmentation methods, we compared measurements of cells (n = 500) and nuclei (n = 400) segmented manually and automatically. Segmentation quality was quantified by the Rand index (RI), aggregated Jaccard index (IoU) and Dice coefficient (DC) (Zou, Warfield et al. 2004, Coelho, Shariff et al. 2009, Stegmaier, Amat et al. 2016, Kumar, Verma et al. 2017, Stringer, Wang et al. 2021) (Figure L4). The average IoU, RI, and DC were 0.83, 0.92, and 0.90 for cells and 0.71, 0.84, and 0.82 for nuclei. These values are largely in agreement with segmentations of good quality (Stegmaier, Amat et al. 2016, Kumar, Verma et al. 2017, Stringer, Wang et al. 2021).

Because we used nuclei only to filter the cell data and measure density in the growth plates, we continued the segmentation quality analysis only on cells.

Figure L4. Quantification of segmentation quality. Segmentation quality was assessed using the Rand index (RI), aggregated Jaccard index (IoU) and Dice coefficient (DC) on cells or nuclei segmented both manually and automatically. The mean IoU, RI, and DC were 0.83, 0.92, and 0.90 for cells (n = 500) and 0.71, 0.84, and 0.82 for nuclei (n = 400).

To evaluate the impact of differences between the two segmentation methods, we computed the percentage deviation (segmentation error) of several morphological characteristics (volume, surface area, sphericity, PC1/2/3 coefficient and orientation) per growth plate zone (Figure L5), as well as the Bland-Altman test of agreement (Martin Bland and Altman 1986), as suggested by the reviewer (Figure L6). The mean percentage deviation per zone for each cell measurement was less than the biological differences we observed, suggesting that the segmentation error is below the resolution of the described phenomena. For example, we observed that cells changed their volume 20% from the RZ to the PZ, 70% from the PZ to the PHZ, and 50% from the PHZ and HZ, while the mean volume segmentation error was 15.13% in the RZ, 12.32% in the PZ, 27.07% in the PHZ and 26.54% in the HZ. Likewise, we observed an 8% increase in cell sphericity in *Gdf5* KO RZ cells, while the segmentation error was 4.09%. Based on these new data, we revised our statement in the manuscript that the *Gdf5* KO RZ cells have 10% larger volumes than controls. The Bland-Altman test of agreement showed similar results to the percentage deviation graphs. Notably, it showed the spread of the segmentation errors from the mean and differences in segmentation quality between zones. For example, measurements of RZ and PZ cells had the lowest segmentation errors, while those of HZ cells had the largest errors.

Figure L5. Percentage deviation between manual and automated segmentations per zone. The percentage deviation was calculated for cell volume (a), surface area (b), sphericity (c), PC1 coefficient (d), PC2 coefficient (e), PC3 coefficient (f), PC1 orientation (g), PC2 orientation (h), and PC3 orientation (i). $n_{rz} = 200$, $n_{pz} = 100$, $n_{phz} = 100$, $n_{hz} = 100$.

Figure L6. Bland-Altman test of agreement between manual and automated segmentations per zone. The test was performed for cell volume, surface area, sphericity, PC1 coefficient, PC2 coefficient, PC3 coefficient. $n_{rz} = 200$, $n_{pz} = 100$, $n_{phz} = 100$, $n_{hz} = 100$. The bold horizontal line on each graph denotes the mean of the data and the two dashed lines denote 2σ , indicating that 95% of the data falls between them.

4/ Finally, in scientific papers, the authors generally acknowledge and discuss the limitations of their study, and this part is missing : for example the sample size is quite limited, is it sufficient to draw conclusions as regards the difference *Gdf5* KO tibiae to controls, given the variability among individuals ?

As the reviewer suggested, we added a Discussion paragraph on the limitations of our study. Regarding our conclusions about *Gdf5* KO tibiae, our pipeline provided a cellular resolution of $80\ \mu\text{m}^3$ in volume, $50\ \mu\text{m}^2$ in surface area, 0.02 in sphericity, 2.2 in PC1 coefficient strength, 0.75 in PC2 coefficient strength, 0.26 in PC3 coefficient strength, 6° in PC1 orientation, 7° in PC2 orientation, and 2° in PC3 orientation. At these resolutions and with the range of segmentation errors, 3D MAPs was sensitive enough to identify differences in all parameters in the *Gdf5* KO bones despite the low sample size of 4 mutants and 5 controls.

References

- Armour, W. J., D. A. Barton, A. M. K. Law and R. L. Overall (2015). "Differential Growth in Periclinal and Anticlinal Walls during Lobe Formation in Arabidopsis Cotyledon Pavement Cells." *The Plant Cell* **27**(9): 2484.
- Bénazéraf, B., M. Beaupeux, M. Tchernookov, A. Wallingford, T. Salisbury, A. Shirtz, A. Shirtz, D. Huss, O. Pourquié, P. François and R. Lansford (2017). "Multi-scale quantification of tissue behavior during amniote embryo axis elongation." *Development* **144**(23): 4462.
- Bénazéraf, B., P. François, R. E. Baker, N. Denans, C. D. Little and O. Pourquié (2010). "A random cell motility gradient downstream of FGF controls elongation of an amniote embryo." *Nature* **466**(7303): 248-252.
- Blanc, P., K. Coste, P. Pouchin, J.-M. Azaïs, L. Blanchon, D. Gallot and V. Sapin (2012). "A Role for Mesenchyme Dynamics in Mouse Lung Branching Morphogenesis." *PLOS ONE* **7**(7): e41643.
- Breur, G. J., L. Md., K. K., S. Km. and G. P. McCabe (1997). "The domain of hypertrophic chondrocytes in growth plates growing at different rates." (0171-967X (Print)).
- Breur, G. J., B. A. Vanenkevort, C. E. Farnum and N. J. Wilsman (1991). "Linear relationship between the volume of hypertrophic chondrocytes and the rate of longitudinal bone growth in growth plates." *Journal of Orthopaedic Research* **9**(3): 348-359.
- Capellini, T. D., H. Chen, J. Cao, A. C. Doxey, A. M. Kiapour, M. Schoor and D. M. Kingsley (2017). "Ancient selection for derived alleles at a GDF5 enhancer influencing human growth and osteoarthritis risk." *Nature genetics* **49**(8): 1202-1210.
- Carter, R., Y. E. Sánchez-Corrales, M. Hartley, V. A. Grieneisen and A. F. M. Marée (2017). "Pavement cells and the topology puzzle." *Development* **144**(23): 4386.
- Chen, H., T. D. Capellini, M. Schoor, D. P. Mortlock, A. H. Reddi and D. M. Kingsley (2016). "Heads, Shoulders, Elbows, Knees, and Toes: Modular Gdf5 Enhancers Control Different Joints in the Vertebrate Skeleton." *PLoS genetics* **12**(11): e1006454-e1006454.
- Coelho, L. P., A. Shariff and R. F. Murphy (2009). "NUCLEAR SEGMENTATION IN MICROSCOPE CELL IMAGES: A HAND-SEGMENTED DATASET AND COMPARISON OF ALGORITHMS." *Proceedings. IEEE International Symposium on Biomedical Imaging* **5193098**: 518-521.
- Cooper, K. L. (2019). "Developmental and Evolutionary Allometry of the Mammalian Limb Skeleton." *Integrative and Comparative Biology* **59**(5): 1356-1368.
- Cooper, K. L., S. Oh, Y. Sung, R. R. Dasari, M. W. Kirschner and C. J. Tabin (2013). "Multiple Phases of Chondrocyte Enlargement Underlie Differences in Skeletal Proportions." *Nature* **495**(7441): 375-378.
- Farhadifar, R., J.-C. Röper, B. Aigouy, S. Eaton and F. Jülicher (2007). "The Influence of Cell Mechanics, Cell-Cell Interactions, and Proliferation on Epithelial Packing." *Current Biology* **17**(24): 2095-2104.
- Farnum, C. E., R. Lee, K. O'Hara and J. P. G. Urban (2002). "Volume increase in growth plate chondrocytes during hypertrophy: the contribution of organic osmolytes." *Bone* **30**(4): 574-581.
- Fu, Y., Y. Gu, Z. Zheng, G. Wasteneys and Z. Yang (2005). "Arabidopsis Interdigitating Cell Growth Requires Two Antagonistic Pathways with Opposing Action on Cell Morphogenesis." *Cell* **120**(5): 687-700.
- HA., S. (1955). Experimental study on the effect of local irritation on bone growth. *Progress in Radiobiology, Proceedings of the Fourth International Conference on Radiobiology*. B. F. S. C. L. Mitchell JSH, editor. Edinburgh and London, Oliver & Boyd: 436-448.
- Hogan, B. L. M. (1999). "Morphogenesis." *Cell* **96**(2): 225-233.
- Irvine, K. D. and E. Wieschaus (1994). "Cell intercalation during Drosophila germband extension and its regulation by pair-rule segmentation genes." *Development* **120**(4): 827.
- Kelvin, L. (1891). Electrical Units of Measurement. *Popular Lectures and Addresses*. **1**.

Kember, N. F. and K. V. R. Walker (1971). "Control of Bone Growth in Rats." *Nature* **229**(5284): 428-429.

Kim, H. Y. and L. A. Davidson (2011). "Punctuated actin contractions during convergent extension and their permissive regulation by the non-canonical Wnt-signaling pathway." *Journal of Cell Science* **124**(4): 635.

Kumar, N., R. Verma, S. Sharma, S. Bhargava, A. Vahadane and A. Sethi (2017). "A Dataset and a Technique for Generalized Nuclear Segmentation for Computational Pathology." *IEEE Transactions on Medical Imaging* **36**(7): 1550-1560.

Lawson, A. and G. C. Schoenwolf (2001). "New insights into critical events of avian gastrulation." *Anat Rec*(0003-276X (Print)).

Lecuit, T. and L. Le Goff (2007). "Orchestrating size and shape during morphogenesis." *Nature* **450**(7167): 189-192.

Lecuit, T. and P.-F. Lenne (2007). "Cell surface mechanics and the control of cell shape, tissue patterns and morphogenesis." *Nature Reviews Molecular Cell Biology* **8**(8): 633-644.

Lecuit, T. and L. Mahadevan (2017). "Morphogenesis one century after &em&em;On Growth and Form&em&em;." *Development* **144**(23): 4197.

Lefevre, J. G., K. M. Short, T. O. Lamberton, O. Michos, D. Graf, I. M. Smyth and N. A. Hamilton (2017). "Branching morphogenesis in the developing kidney is governed by rules that pattern the ureteric tree." *Development* **144**(23): 4377.

Lewis, F. T. (1931). "A comparison between the mosaic of polygons in a film of artificial emulsion and the pattern of simple epithelium in surface view (cucumber epidermis and human amnion)." *The Anatomical Record* **50**(3): 235-265.

Li, Y., V. Trivedi, T. V. Truong, D. S. Koos, R. Lansford, C.-M. Chuong, D. Warburton, R. A. Moats and S. E. Fraser (2015). "Dynamic imaging of the growth plate cartilage reveals multiple contributors to skeletal morphogenesis." *Nature Communications* **6**: 6798.

Lui, J. C., Y. H. Jee, P. Garrison, J. R. Iben, S. Yue, M. Ad, Q. Nguyen, B. Kikani, Y. Wakabayashi and J. Baron (2018). "Differential aging of growth plate cartilage underlies differences in bone length and thus helps determine skeletal proportions." *PLOS Biology* **16**(7): e2005263.

Martin, A. C., M. Kaschube and E. F. Wieschaus (2009). "Pulsed contractions of an actin–myosin network drive apical constriction." *Nature* **457**(7228): 495-499.

Martin Bland, J. and D. Altman (1986). "STATISTICAL METHODS FOR ASSESSING AGREEMENT BETWEEN TWO METHODS OF CLINICAL MEASUREMENT." *The Lancet* **327**(8476): 307-310.

Metzger, R. J., O. D. Klein, G. R. Martin and M. A. Krasnow (2008). "The branching programme of mouse lung development." *Nature* **453**(7196): 745-750.

Mikic, B., C. Rt, T. C. Battaglia, V. Gaschen and E. B. Hunziker (2004). "Altered hypertrophic chondrocyte kinetics in GDF-5 deficient murine tibial growth plates." (0736-0266 (Print)).

Newton, P. T., L. Li, B. Zhou, C. Schweingruber, M. Hovorakova, M. Xie, X. Sun, L. Sandhow, A. V. Artemov, E. Ivashkin, S. Suter, V. Dyachuk, M. El Shahawy, A. Gritli-Linde, T. Boudierlique, J. Petersen, A. Mollbrink, J. Lundberg, G. Enikolopov, H. Qian, K. Fried, M. Kasper, E. Hedlund, I. Adameyko, L. Sävendahl and A. S. Chagin (2019). "A radical switch in clonality reveals a stem cell niche in the epiphyseal growth plate." *Nature* **567**(7747): 234-238.

Paluch, E. and C. P. Heisenberg (2009). "Biology and physics of cell shape changes in development." (1879-0445 (Electronic)).

Settle, S. H., R. B. Rountree, A. Sinha, A. Thacker, K. Higgins and D. M. Kingsley (2003). "Multiple joint and skeletal patterning defects caused by single and double mutations in the mouse Gdf6 and Gdf5 genes." *Developmental Biology* **254**(1): 116-130.

Shih, J. and R. Keller (1992). "Cell motility driving mediolateral intercalation in explants of *Xenopus laevis*." *Development* **116**(4): 901.

Shih, J. and R. Keller (1992). "Patterns of cell motility in the organizer and dorsal mesoderm of *Xenopus laevis*." Development **116**(4): 915.

Short, Kieran M., Alexander N. Combes, J. Lefevre, Adler L. Ju, Kylie M. Georgas, T. Lamberton, O. Cairncross, Bree A. Rumballe, Andrew P. McMahon, Nicholas A. Hamilton, Ian M. Smyth and Melissa H. Little (2014). "Global Quantification of Tissue Dynamics in the Developing Mouse Kidney." Developmental Cell **29**(2): 188-202.

Shwartz, Y., S. Viukov, S. Krief and E. Zelzer (2016). "Joint Development Involves a Continuous Influx of Gdf5-Positive Cells." (2211-1247 (Electronic)).

Stegmaier, J., F. Amat, William C. Lemon, K. McDole, Y. Wan, G. Teodoro, R. Mikut and Philipp J. Keller (2016). "Real-Time Three-Dimensional Cell Segmentation in Large-Scale Microscopy Data of Developing Embryos." Developmental Cell **36**(2): 225-240.

Storm, E. E. and D. M. Kingsley (1996). "Joint patterning defects caused by single and double mutations in members of the bone morphogenetic protein (BMP) family." Development **122**(12): 3969.

Storm, E. E., H. T.V., C. N.G., J. N.A., K. D.M. and S. J. Lee (1994). "Limb alterations in brachypodism mice due to mutations in a new member of the TGF beta-superfamily." (0028-0836 (Print)).

Stringer, C., T. Wang, M. Michaelos and M. Pachitariu (2021). "Cellpose: a generalist algorithm for cellular segmentation." Nature Methods **18**(1): 100-106.

Warga, R. M. and C. B. Kimmel (1990). "Cell movements during epiboly and gastrulation in zebrafish." Development **108**(4): 569.

Wilsman, N. J., E. S. Bernardini, E. Leiferman, K. Noonan and C. E. Farnum (2008). "Age and Pattern of the Onset of Differential Growth Among Growth Plates in Rats." Journal of orthopaedic research : official publication of the Orthopaedic Research Society **26**(11): 1457-1465.

Wilsman, N. J., C. Farnum, E. Leiferman, M. Fry and C. Barreto (1996). "Differential growth by growth plates as a function of multiple parameters of chondrocytic kinetics." (0736-0266 (Print)).

Zou, K. H., S. K. Warfield, A. Bharatha, C. M. C. Tempany, M. R. Kaus, S. J. Haker, W. M. Wells, 3rd, F. A. Jolesz and R. Kikinis (2004). "Statistical validation of image segmentation quality based on a spatial overlap index." Academic radiology **11**(2): 178-189.

REVIEWER COMMENTS

Reviewer #1 (Remarks to the Author):

I will retain the comment numbers from the original review.

1. The abstract now spells out the acronym but still does not explain what 3D MAPs is. The reader cannot really understand the abstract without a brief (e.g. one-sentence) description of the method. The term "3D Morphometric Analysis for Phenotypic significance" does not provide that information.

6. The authors' response does not substantively address the original comment (reiterated here): The results subsection entitled, "Allometric and isometric growth behaviors define chondrocyte shape throughout differentiation," contains little novel information. It has been known for decades that the resting zone chondrocytes are something like spheres, the proliferative zone chondrocytes are flattened like pancakes, and the hypertrophic zone chondrocytes lose that flattening. Thus, the diameter parallel to the proximal-distal axis of the bone decreases than increases. This subsection seems to restate this well-known concept using the terms isometric and allometric, which, in my view, is an unnecessarily convoluted way to view the straightforward geometric changes. Of course, a sphere will have a greater volume vs surface area than a short cylinder (pancake), but the changes are far easier to conceptualize in terms of simple changes in shape than in terms of $\text{Vol}^{2/3} / \text{SA}$. Thus the subsection has two problems. First it contains little new information, and second it obfuscates.

The paper would be improved by omitting (or at least minimizing) the unnecessarily convoluted analysis in terms of $\text{Vol}^{2/3} / \text{Area}$ and omitting the terms "allometric" and "isometric." Instead, the findings could be interpreted in the simple, intuitive concept of growth along the proximal-distal axis versus other axes. Indeed, the paragraph which begins, "To reveal the specific mechanism..." does this well.

It is also important to acknowledge in the paper that the fundamental concept of shape change (spheroid to discoid to spheroid) has been known for many years and therefore this new method did not discover a new concept but only provides a more detailed quantitative assessment of shape change as cells progress from the proximal to distal growth plate.

7. The study found that growth plate chondrocytes are longer in the dorsal-ventral axis than in the medial-lateral axis. The original reviewer comment raised the concern that this asymmetry in the medial-lateral diameter vs the dorsal-ventral diameter might be a false result arising from the asymmetry in the measurement approach. To address this concern the authors repeated the imaging at two orthogonal rotations along the P-D axis, such that the z axis was along either the dorsal-ventral or medial-lateral axis of the bone. This information is reassuring. I suggest that this additional information, including Fig. L2, be included in the published article (e.g. mentioned very briefly in the main article with details in the supplemental materials), not just the response to authors.

8. The authors have now studied Gdf5 expression using single-molecule fluorescent in situ hybridization. It would be important to have a negative control, preferably the knockout mouse, if it lacks the amplified region or entire mRNA.

The authors also now show pSMAD 1/5/9 phosphorylation by immunofluorescence. An analysis of SMAD phosphorylation is particularly valuable because it is not clear that the very low level of Gdf5 expression is biologically significant. However, immunofluorescence is not a quantitative method. Fluorescence can vary considerably even when done on two samples that should be biologically identical. Comparisons are therefore only convincing when the difference is striking, which is not the case here, and reproducible. At a minimum, I would suggest having slides (without prior selection) scored for staining by an observer blinded to genotype. The two sets of slides (WT and knockout) should have been sectioned, processed, and stained concurrently. The number of biological replicates (mice) should be sufficient to provide adequate power for the comparison.

9. The claims made in the paper about how this method provides mechanistic insights seem overly enthusiastic based on the data presented. For example, the method allows a description of the cell size and shape abnormalities in Gdf5 knockout mice, but it did not provide much insight into the underlying chain of molecular mechanisms that lie downstream from Gdf5 binding to receptors eventually resulting in abnormal chondrocyte differentiation. Therefore, I suggest toning down some of the claims made in this paper. For example, I suggest modifying the statement in the abstract that, "Overall, our findings provide new insight into the morphological sequence that chondrocytes undergo during differentiation [well supported by the data] and highlight the ability

of 3D MAPs to uncover molecular [not supported by the data] and cellular mechanisms [the study was unable to penetrate beyond the changes in cell shape using this technique, except to suggest that some unknown processes are occurring all zones] regulating this process." Similarly, I would omit the concluding sentence of the discussion, "This new pipeline has the potential to change the way we study growth plate biology." The findings presented in this paper support the idea that 3D MAPs is one useful tool but not the idea that it will revolutionize the field.

Reviewer #2 (Remarks to the Author):

The authors adequately responded to the concerns

1. The abstract now spells out the acronym but still does not explain what 3D MAPs is. The reader cannot really understand the abstract without a brief (e.g. one-sentence) description of the method. The term “3D Morphometric Analysis for Phenotypic significance” does not provide that information.

As the reviewer requested, we have added the following description of 3D MAPs in the abstract:

“To overcome these obstacles, we have developed a pipeline called 3D Morphometric Analysis for Phenotypic significance (3D MAPs), which combines light-sheet microscopy, segmentation algorithms and 3D morphometric analysis to characterize morphogenetic cellular behaviors while maintaining the spatial context of the tissue.”

6. The authors’ response does not substantively address the original comment (reiterated here):

The results subsection entitled, “Allometric and isometric growth behaviors define chondrocyte shape throughout differentiation,” contains little novel information. It has been known for decades that the resting zone chondrocytes are something like spheres, the proliferative zone chondrocytes are flattened like pancakes, and the hypertrophic zone chondrocytes lose that flattening. Thus, the diameter parallel to the proximal-distal axis of the bone decreases than increases. This subsection seems to restate this well-known concept using the terms isometric and allometric, which, in my view, is an unnecessarily convoluted way to view the straightforward geometric changes. Of course, a sphere will have a greater volume vs surface area than a short cylinder (pancake), but the changes are far easier to conceptualize in terms of simple changes in shape than in terms of $Vol^{2/3} / SA$. Thus the subsection has two problems. First it contains little new information, and second it obfuscates.

The paper would be improved by omitting (or at least minimizing) the unnecessarily convoluted analysis in terms of $Vol^{2/3}/Area$ and omitting the terms “allometric” and “isometric.” Instead, the findings could be interpreted in the simple, intuitive concept of growth along the proximal-distal axis versus other axes. Indeed, the paragraph which begins, “To reveal the specific mechanism...” does this well.

It is also important to acknowledge in the paper that the fundamental concept of shape change (spheroid to discoid to spheroid) has been known for many years and therefore this new method did not discover a new concept but only provides a more detailed quantitative assessment of shape change as cells progress from the proximal to distal growth plate.

As suggested by the reviewer, we have clarified that it is known from histological studies that cells in the growth plate start as round in the resting zone, then flatten in the proliferative zone, and become more round again in the hypertrophic zone. We also highlighted in the text that these are qualitative assessments and very few papers have

actually quantified this phenomenon of changes in sphericity in 3D. In the Results section, we clarified that we are the first to describe the transition between morphological states, emphasizing the importance of using the terms isometric and allometric to describe the growth process. We have also added references to the works that inspired us to use the $\text{vol}^{2/3}/\text{sa}$ measurement to describe cell growth, namely Harris and Theriot (2018) and Ojkic and Serbanescu et al. (2019). Below is an excerpt from the modified Results section:

“Histological studies have firmly established that during chondrocyte differentiation, cells change their morphology from round in the resting zone to flattened in the proliferative zone and become more round again in the hypertrophic zone, yet how differentiating chondrocytes transition between morphologies is still unknown. (Dodds 1930, Pacifici, Golden et al. 1990, Abad, Meyers et al. 2002, Amini, Veilleux et al. 2010, Amini, D. et al. 2011). Chondrocyte morphogenesis can utilize different growth mechanisms to achieve the same morphology implicating different cellular machineries, and thus understanding these transitions is fundamental. One way to change cell morphology is by isometric growth, where only cell size changes. The other way is allometric growth, where both size and shape change. Mathematically, isometric growth occurs when the surface area scales to the two-third power compared to the volume (McMahon, Bonner et al. 1983, Okie 2013, Harris and Theriot 2018, Ojkic, Serbanescu et al. 2019), while in allometric growth, this scaling factor differs.”

7. The study found that growth plate chondrocytes are longer in the dorsal-ventral axis than in the medial-lateral axis. The original reviewer comment raised the concern that this asymmetry in the medial-lateral diameter vs the dorsal-ventral diameter might be a false result arising from the asymmetry in the measurement approach. To address this concern the authors repeated the imaging at two orthogonal rotations along the P-D axis, such that the z axis was along either the dorsal-ventral or medial-lateral axis of the bone. This information is reassuring. I suggest that this additional information, including Fig. L2, be included in the published article (e.g. mentioned very briefly in the main article with details in the supplemental materials), not just the response to authors.

As the reviewer requested, we included the figure disproving an imaging artifact into the Supplementary section of the manuscript, with the following references made in the main text:

“To verify that the imaging angle does not introduce an artifact, we analyzed an ulna imaged at two orthogonal rotations. Results showed that the orientation of PC1 was not changed between the two imaging angles, which was further validated quantitatively by performing a segmentation error analysis (Supplementary Figures 2g and 5), excluding the possibility of an artifact.”

8. The authors have now studied Gdf5 expression using single-molecule fluorescent in situ hybridization. It would be important to have a negative control, preferably the knockout mouse, if it lacks the amplified region or entire mRNA.

The authors also now show pSMAD 1/5/9 phosphorylation by immunofluorescence. An analysis of SMAD phosphorylation is particularly valuable because it is not clear that the very low level of Gdf5 expression is biologically significant. However, immunofluorescence is not a quantitative method. Fluorescence can vary considerably even when done on two samples that should be biologically identical. Comparisons are therefore only convincing when the difference is striking, which is not the case here, and reproducible. At a minimum, I would suggest having slides (without prior selection) scored for staining by an observer blinded to genotype. The two sets of slides (WT and knockout) should have been sectioned, processed, and stained concurrently. The number of biological replicates (mice) should be sufficient to provide adequate power for the comparison.

Following the reviewer's suggestion, we included a control for the HCR smFISH. The *Gdf5* KO mice were engineered by introducing a CreER^{T2} cassette in frame into the ATG of the *Gdf5* gene, without excision of the original exons (Shwartz, Viukov et al. 2016). Some of the RNA production by this locus is maintained, preventing the *Gdf5* KO from serving as a negative control for the in situ. Instead, we used samples consisting of only the amplifier plus fluorophore as the negative control, as suggested by the manufacturer of the HCR probes (www.molecularinstruments.com). As seen in Supplementary Figure 6A, there is no signal in the negative control, demonstrating the validity of our findings.

To address the reviewer's concerns about the robustness of the reduced SMAD activity in the *Gdf5* KO mice, we performed four different staining experiments on three different mouse litters. To accommodate the request for a blind test, we imaged all of the sections from each slide and masked the growth plate. Then we used the Fiji (Schindelin, Arganda-Carreras et al. 2012) "Find Maxima" tool to count the number of nuclei in each growth plate and thresholding/segmentation tools to count the number of nuclear pSmad 1/5/9 cells. We next calculated the ratio of nuclear pSmad 1/5/9 cells per growth plate by dividing the number of nuclear pSmad 1/5/9 by the total number of nuclei in the growth plate (see Supplementary Methods). The data are presented as a box and whiskers plot in Supplementary Figure 6B, and a two-way ANOVA test with an alpha of 0.05 showed that the *Gdf5* KO growth plates have significantly less nuclear pSmad 1/5/9 than control growth plates.

9. The claims made in the paper about how this method provides mechanistic insights seem overly enthusiastic based on the data presented. For example, the method allows a description of the cell size and shape abnormalities in *Gdf5* knockout mice, but it did not provide much insight into the underlying chain of molecular mechanisms that lie downstream from *Gdf5* binding to receptors eventually resulting in abnormal chondrocyte differentiation. Therefore, I suggest toning down some of the claims made in this paper. For example, I suggest modifying the statement in the abstract that, "Overall, our findings provide new insight into the morphological sequence that chondrocytes undergo during differentiation [well supported by the data] and highlight the ability of 3D MAPs to uncover molecular [not supported by the data] and cellular mechanisms [the study was unable to penetrate beyond the changes in cell shape using this technique, except to suggest that some unknown processes are

occurring all zones] regulating this process.” Similarly, I would omit the concluding sentence of the discussion, “This new pipeline has the potential to change the way we study growth plate biology.” The findings presented in this paper support the idea that 3D MAPs is one useful tool but not the idea that it will revolutionize the field.

As requested by the reviewer, to tone down the claims we modified the sentence in the abstract to the following:

“Overall, our findings provide new insight into the morphological sequence that chondrocytes undergo during differentiation and highlight the ability of 3D MAPs to uncover cellular mechanisms that may regulate this process.”

Additionally, as requested we omitted the last sentence of the Discussion.

REVIEWERS' COMMENTS

Reviewer #1 (Remarks to the Author):

The authors' response still does not substantively address Reviewer Comment 6 (reiterated here): The results subsection entitled, "Allometric and isometric growth behaviors define chondrocyte shape throughout differentiation," contains little novel information. It has been known for decades that the resting zone chondrocytes are something like spheres, the proliferative zone chondrocytes are flattened like pancakes, and the hypertrophic zone chondrocytes lose that flattening. Thus, the diameter parallel to the proximal-distal axis of the bone decreases then increases. This subsection seems to restate this well-known concept using the terms isometric and allometric, which, in my view, is an unnecessarily convoluted way to view the straightforward geometric changes. Of course, a sphere will have a greater volume vs surface area than a short cylinder (pancake), but the changes are far easier to conceptualize in terms of simple changes in shape than in terms of $\text{Vol}^{2/3} / \text{SA}$. Thus the subsection has two problems. First it contains little new information, and second it obfuscates.

The paper would be improved by omitting (or at least minimizing) the unnecessarily convoluted analysis in terms of $\text{Vol}^{2/3}/\text{Area}$ and omitting the terms "allometric" and "isometric." Instead, the findings could be interpreted in the simple, intuitive concept of growth along the proximal-distal axis versus other axes. Indeed, the paragraph which begins, "To reveal the specific mechanism..." does this well. It is also important to acknowledge in the paper that the fundamental concept of shape change (spheroid to discoid to spheroid) has been known for many years and therefore this new method did not discover a new concept but only provides a more detailed quantitative assessment of shape change as cells progress from the proximal to distal growth plate.

Thus, in my opinion, this section only restates a well-understood concept in an unnecessarily convoluted form and does not advance the field. However, it is not overtly wrong. I would defer to the editor as to who is correct here and whether it is worth insisting.

Reviewer #1 (Remarks to the Author):

The authors' response still does not substantively address Reviewer Comment 6 (reiterated here): The results subsection entitled, "Allometric and isometric growth behaviors define chondrocyte shape throughout differentiation," contains little novel information. It has been known for decades that the resting zone chondrocytes are something like spheres, the proliferative zone chondrocytes are flattened like pancakes, and the hypertrophic zone chondrocytes lose that flattening. Thus, the diameter parallel to the proximal-distal axis of the bone decreases than increases. This subsection seems to restate this well-known concept using the terms isometric and allometric, which, in my view, is an unnecessarily convoluted way to view the straightforward geometric changes. Of course, a sphere will have a greater volume vs surface area than a short cylinder (pancake), but the changes are far easier to conceptualize in terms of simple changes in shape than in terms of $Vol^{2/3} / SA$.

Thus the subsection has two problems. First it contains little new information, and second it obfuscates.

The paper would be improved by omitting (or at least minimizing) the unnecessarily convoluted analysis in terms of $Vol^{2/3}/Area$ and omitting the terms "allometric" and "isometric." Instead, the findings could be interpreted in the simple, intuitive concept of growth along the proximal-distal axis versus other axes. Indeed, the paragraph which begins, "To reveal the specific mechanism..." does this well. It is also important to acknowledge in the paper that the fundamental concept of shape change (spheroid to discoid to spheroid) has been known for many years and therefore this new method did not discover a new concept but only provides a more detailed quantitative assessment of shape change as cells progress from the proximal to distal growth plate.

Thus, in my opinion, this section only restates a well-understood concept in an unnecessarily convoluted form and does not advance the field. However, it is not overtly wrong. I would defer to the editor as to who is correct here and whether it is worth insisting.

As the reviewer states, we have a difference of opinions about the importance of the terminology and findings of this subsection. We defer to the editor's opinion.